# Computing Circuits Optimization via Model-Based Circuit Genetic Evolution

**Zhihai Wang[1]**[*]**, Jie Wang[1]**[†]**, Xilin Xia[1], Dongsheng Zuo[3], Lei Chen[2], Yuzhe Ma[3], JianYe Hao[2,4], Mingxuan Yuan[2], Feng Wu[1]**

[1]MoE Key Laboratory of Brain-inspired Intelligent Perception and Cognition, University of Science and Technology of China
[2] Noah's Ark Lab, Huawei Technologies
[3] Microelectronics Thrust, Hong Kong University of Science and Technology (Guangzhou)
[4] College of Intelligence and Computing, Tianjin University

## Abstract

Optimizing computing circuits such as multipliers and adders is a fundamental challenge in modern integrated circuit design. Recent efforts propose formulating this optimization problem as a reinforcement learning (RL) proxy task, offering a promising approach to search high-speed and area-efficient circuit design solutions. However, we show that the RL-based formulation (proxy task) converges to a *local optimal* design solution (original task) due to the deceptive reward signals and incrementally localized actions in the RL-based formulation. To address this challenge, we propose a novel **m**odel-based circ**u**it gene**t**ic **e**volution (MUTE) framework, which reformulates the problem as a genetic evolution process by proposing a grid-based genetic representation of design solutions. This novel formulation avoids misleading rewards by evaluating and improving generated solutions using the true objective value rather than proxy rewards. To promote globally diverse exploration, MUTE proposes a *multi-granularity* genetic crossover operator that recombines design substructures at varying column ranges between two grid-based genetic solutions. To the best of our knowledge, MUTE is *the first* to reformulate the problem as a circuit genetic evolution process, which enables effectively searching for global optimal design solutions. We evaluate MUTE on several fundamental computing circuits, including multipliers, adders, and multiply-accumulate circuits. Experiments on these circuits demonstrate that MUTE significantly Pareto-dominates state-of-the-art approaches in terms of both area and delay. Moreover, experiments demonstrate that circuits designed by MUTE well generalize to large-scale computation-intensive circuits as well.

## 1 Introduction

Computing circuits such as multipliers and adders serve as the fundamental building blocks in numerous real-world circuits, particularly in central processing units, graphics processing units, and artificial intelligence (AI) chips (Holdsworth, 1987; Das et al., 2019; Sze et al., 2020). The multiplication and addition operations stand out as the most fundamental and frequently utilized arithmetic operations across various computation-intensive applications, including deep neural networks (DNNs), digital signal processors, and microprocessors (Hashemian, 2002; Elguibaly, 2000; Zuo et al., 2023). Notably, in many popular DNN architectures such as ResNet (He et al., 2016), ViT (Dosovitskiy et al., 2021), Transformer (Vaswani et al., 2017), and BERT (Devlin et al., 2019), the multiplication and addition operations constitute over 99% of all operations. Therefore, the design of high-speed and area-efficient computing circuits plays a pivotal role in enhancing the performance of computation-intensive applications, especially in AI chips.

However, computing circuit optimization is a challenging combinatorial optimization problem due to its $\mathcal{NP}$-hard nature (Hillar & Lim, 2013; Song et al., 2022). On one hand, the combinatorial

---

[*]This work was done when Zhihai Wang was an intern at Huawei Noah's Ark Lab.
[†]Corresponding author. Email: jiewangx@ustc.edu.cn.

design space grows exponentially with the input bit widths of the computing circuits (Roy et al., 2021). On the other hand, evaluating the post-synthesis performance of a circuit design (i.e., design performance) with circuit synthesis tools is highly time-consuming, leading to high sampling costs. Therefore, searching high-speed and area-efficient circuits in the vast design space using limited samples emerges as a significant challenge.

To search high-speed and area-efficient circuits, recent efforts (Roy et al., 2021; Zuo et al., 2023; Song et al., 2022) propose formulating the computing circuits optimization problem as a reinforcement learning (RL) proxy task, offering a promising avenue for optimizing circuit designs using limited samples. Specifically, they start from an initial design solution, learn policies to incrementally modify the local design structure, and utilize design performance gains between two consecutive designs as reward signals. Intuitively, the manually designed rewards can guide RL agents to explore directions that progressively improve design performance at each step.

However, we show that the RL-based formulation (proxy task) converges to a *local optimal* design solution (original task) due to the deceptive reward signals and incrementally localized actions. First, the reward signals based on performance gains between two consecutive designs are deceptive, as maximizing the cumulative discounted rewards misaligns with the true objective. More specifically, the proxy RL formulation indeed optimizes the cumulative discounted performance of all encountered design solutions across a trajectory, while the true objective is to find the single best-performing designs. Second, the actions based on the incrementally local modifications of design structure suffer from poor exploration capability, and thus struggle to escape local optima.

To address these challenges, we propose a novel **m**odel-based circ**u**it gene**t**ic **e**volution (MUTE) framework, which proposes a grid-based genetic representation of solutions and reformulates the problem as a circuit genetic evolution process. The evolution formulation is an iterative process between circuit genetic variation and model-based selection, where each iteration evaluates and improves solutions using the true objective value, thus gradually converging toward the best-performing solution (i.e., the original task). To promote globally diverse exploration for escaping local optima, MUTE proposes a multi-granularity crossover operator that recombines design substructures at varying column ranges between two grid-based genetic solutions. Moreover, to tackle the problem of high sampling costs, MUTE introduces a model-based selection method, which learns a model for rapid evaluation of a large number of solutions.

We evaluate MUTE on several fundamental computing circuits, including multipliers, adders, and multiply-accumulate circuits. Experiments on these circuits, spanning a wide range of input widths, demonstrate that MUTE discovers state-of-the-art designs that significantly Pareto-dominate those produced by manual design, mathematical optimization, and learning-based approaches, improving the hypervolume by up to 38%. Moreover, we deploy circuits optimized by MUTE and the baselines into large-scale computation-intensive circuits, and experiments show that MUTE significantly outperforms the baselines in terms of both area and delay. Our results highlight the superior ability of MUTE to discover high-speed and area-efficient circuits for real-world important computing applications, especially for high-performance AI chips.

We summarize our major contributions as follows. (1) We show that the RL-based formulation for computing circuits optimization converges to a local optimal design solution, indicating a significant objective gap between the RL-based formulation and the true objective. (2) To the best of our knowledge, our MUTE is *the first* to reformulate the optimization problem as a novel circuit genetic evolution process, which enables effectively searching for the global optimal circuit design solutions. (3) MUTE proposes a multi-granularity genetic crossover operator to promote globally diverse exploration of the design space. (4) Experiments show that MUTE significantly outperforms state-of-the-art approaches in terms of both area and delay.

## 2 BACKGROUND

### 2.1 COMPUTING CIRCUITS ARCHITECTURE

Most computing circuits such as prefix adders, vector adders, subtracters, multipliers, and multiply-accumulate circuits rely on two fundamental circuit structures, i.e., the Compressor Tree and Prefix Tree (Weste & Harris, 2015; Roy et al., 2021; Zuo et al., 2023; Wang et al., 2024g). Note that the Compressor Tree and Prefix Tree both share similar tree structures that can both be represented by grid-based design solutions. We take a multiplier circuit with four input bits as an example to in-

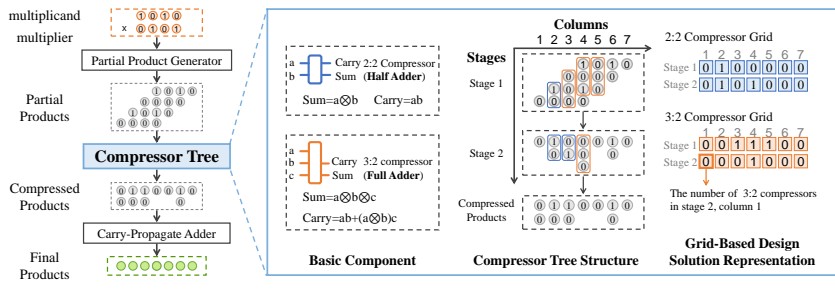

Figure 1: An illustration of the multiplication process and multiplier architecture.

troduce the Compressor Tree structure as shown in Figure 1. In binary multiplication, two unsigned binary numbers—the multiplicand and the multiplier—are combined to yield their product. Contemporary multiplier designs typically comprise three primary components: a partial product generator (PPG), a Compressor Tree, and a carry propagation adder (CPA). **Initially**, the PPG generates a bit matrix based on the multiplicand and multiplier, with each element representing a partial product. **Subsequently**, the Compressor Tree compresses each column of the bit matrix to one or two bits by concurrently summing up the partial products within each column. **Finally**, the CPA aggregates the resultant bit matrix from the Compressor Tree to derive the final product.

In constructing a Compressor Tree, **a large number of full and half adders are typically employed to execute the summation of generated partial products concurrently**. A full adder, i.e., a 3:2 compressor, accepts three inputs—two single-bit values and a carry-in bit—and produces two outputs: a sum bit and a carry-out bit. A half adder, i.e., a 2:2 compressor, takes two single-bit values as inputs and yields two outputs: a sum bit and a carry-out bit. Notably, when a 3:2 (2:2) compressor is applied to the $i$-th column, it reduces two (one) bits in column $i$ while increasing one bit in column $(i + 1)$. Thus, a Compressor Tree employs numerous compressors (i.e., full and half adders) across multiple stages to compress the partial products matrix into only two rows in parallel, significantly dominating the final performance of a multiplier circuit. Moreover, modifying the arrangement of 3:2 and 2:2 compressors within a Compressor Tree can result in significantly different tree structure designs, leading to variable design performance.

## 2.2 RL FOR COMPUTING CIRCUITS OPTIMIZATION

As the Compressor Tree and/or Prefix Tree usually dominates the final performance of a computing circuit (Zuo et al., 2023; Xiao et al., 2021), recent efforts have focused on optimizing the tree structure by formulating the optimization problem as a reinforcement learning (RL) proxy task (Zuo et al., 2023; Roy et al., 2021). We take the existing RL-based Compressor Tree optimization method as an example. RL-MUL starts from an initial Compressor Tree design solution, learns policies to sequentially modify the design structure locally, and utilizes design performance gains as reward signals. We specify the state space, action space, and reward function as follows. **(1) State Space $\mathcal{S}$.** RL-MUL formulates each legal design solution as a state, where each state is represented by a grid-based image. **(2) Action Space $\mathcal{A}$.** RL-MUL designs four types of local modifications to a Compressor Tree solution at a certain column. These local modifications include adding a 2:2 compressor, removing a 2:2 compressor, replacing a 3:2 compressor with a 2:2 compressor, and replacing a 2:2 compressor with a 3:2 compressor. The action space is a discrete set composed of $4 \times N_C$ discrete actions, where $N_C$ denotes the number of columns. Each action $i \in [0, 1, \ldots, (4 \times N_C - 1)]$ is represented by executing the $j$-th modification type at the $k$-th column, where $j = i \pmod 4$ and $k = \lfloor \frac{i}{4} \rfloor$. **(3) Reward Function $r$.** RL-MUL uses a circuit synthesis tool to obtain the performance of the designed solution at each step. The reward $r_t$ is defined as the difference between the area (delay) of the design at step $t - 1$ and that at step $t$. That is, $r(s_t, a_t, s_{t+1}) = f(s_t) - f(s_{t+1})$, where $f$ denotes the design evaluation function. Finally, RL-MUL leverages the deep Q-network algorithm (Mnih et al., 2015) to train Q-networks. We defer details to Appendix D.

# 3 LIMITATIONS OF EXISTING RL FORMULATION

## 3.1 DECEPTIVE REWARD SIGNALS

Existing methods formulate the optimization problem as an infinite-horizon Markov decision process (MDP) denoted by a tuple $(\mathcal{S}, \mathcal{A}, r, T, \gamma, \mu_0)$, where $\mathcal{S}$ denotes the state space, $\mathcal{A}$ denotes the

action space, $r : \mathcal{S} \times \mathcal{A} \times \mathcal{S} \rightarrow \mathbb{R}$ denotes the reward function, $T$ denotes the deterministic transition function, $\gamma$ denotes the discount factor, and $\mu_0$ denotes the given initial design solution. Based on the MDP, the return of a deterministic policy $\pi$ is defined as $R^\pi = \sum_{t=0}^\infty \gamma^t r(s_t, a_t, s_{t+1})$, where $s_0 = \mu_0, a_t = \pi(s_t)$, and $s_{t+1} = T(s_t, a_t)$. Note that the reward is defined by the performance gain between the states $s_t$ and $s_{t+1}$, i.e.,

$$r(s_t, a_t, s_{t+1}) = f(s_t) - f(s_{t+1}). \tag{1}$$

Here $f : \mathcal{S} \rightarrow \mathbb{R}$ denotes the underlying evaluation function of design solutions given by circuit synthesis tools. Note that multiple evaluation functions are employed, such as area and delay evaluation functions. For ease of analysis and consistent with previous work (Roy et al., 2021; Zuo et al., 2023), we assume that the evaluation function $f$ is a linear weighted average of these evaluation functions. Intuitively, the manually designed proxy rewards based on performance gains are able to guide RL agents toward directions that progressively improve design performance, as the RL agent receives positive rewards for improving design performance. Thus, a desired question is: *Does the optimal policy in the RL formulation converge to the global optimal design solution?*

To investigate this question, we first theoretically show *the RL-based optimal policy converges to a local optimal design solution*. Then we empirically show that the underlying evaluation function $f$ is highly oscillatory, resulting in the local optimal design solutions found by the optimal policy diverging significantly from the global optimal solution.

**Theoretical Analysis** We assume that the state space $\mathcal{S}$ is finite. For simplicity, we assume a terminal action for each state that can terminate the episode at this state. We define a state $s \in \mathcal{S}$ as a local optimum of the function $f$ if for all action $a \in \mathcal{A}$ we have $f(T(s, a)) \geq f(s)$.

**Theorem 3.1.** *The optimal RL policy $\pi^* := \arg\max_\pi R^\pi$ terminates at a state, and the state is a local optimal design solution of the evaluation function $f$.*

This theorem demonstrates the superior capability of RL methods in achieving local optimal solutions. However, this raises a further question: *Is the converged local optimal point also the global optimum?* Given the lack of detailed information about the optimization objective function $f$, a rigorous analysis of this problem is currently infeasible. Therefore, we present an intuitive and empirical analysis to demonstrate that *the converged local optimal solution can significantly diverge from the global optimal solution* as follows.

**Illustrative Example** We revisit the optimization objective in the RL formulation, i.e.,

$$R^\pi = \sum_{t=0}^\infty \gamma^t (f(s_t) - f(s_{t+1})) = f(s_0) - \sum_{t=0}^\infty \gamma^t (1 - \gamma) f(s_{t+1}). \tag{2}$$

This implies that standard RL methods in the existing formulation aim to minimize the *cumulative discounted performance* of all visited solutions across a trajectory, except the initial state, when the discount factor $\gamma < 1$. This is a practical discount factor setting in standard RL and previous methods (Roy et al., 2021; Zuo et al., 2023). In contrast, the circuit optimization task is a best-case-seeking task, i.e., the final performance is measured by the *single or few best-performing* design solutions found during training. Consequently, the RL-based optimization objective is inconsistent with the original optimization objective, possibly leading to a significant optimization objective gap.

To illustrate the optimization objective gap problem, we provide a motivating example as shown in Figure 2 (Left). Specifically, we illustrate two distinct trajectories in the circuit optimization environment from a given starting solution following two deterministic policies $\pi_1$ and $\pi_2$. We denote the two trajectories by $(s_0^{\pi_1}, a_0^{\pi_1}, s_1^{\pi_1}, a_1^{\pi_1}, \dots, s_T^{\pi_1})$ and $(s_0^{\pi_2}, a_0^{\pi_2}, s_1^{\pi_2}, a_1^{\pi_2}, \dots, s_T^{\pi_2})$, respectively. Each point in Figure 2

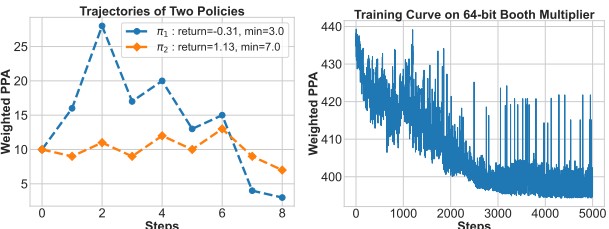

Figure 2: **(Left)** A motivating example of two distinct trajectories with conflicting returns and found best solutions. **(Right)** A practical training curve of the EA method.

(Left) corresponds to the performance of a state across the trajectory. As shown in Figure 2 (Left), the return of the policy $\pi_2$ is larger than that of the policy $\pi_1$, while the best solution found by $\pi_1$

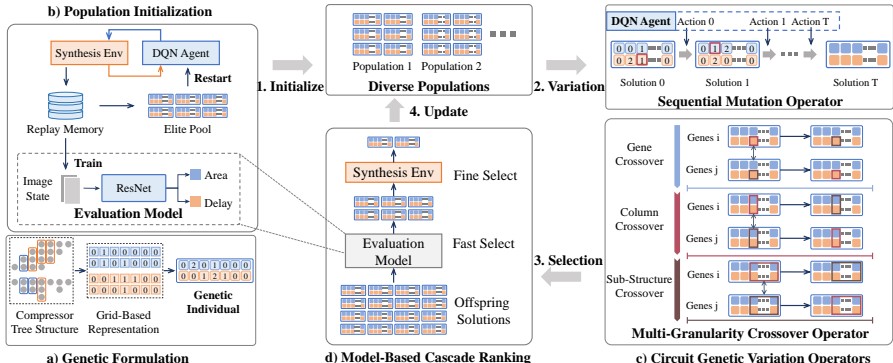

Figure 3: An illustration of the MUTE framework with a) Genetic Formulation, b) Population Initialization, c) Circuit Genetic Variation Operators, and d) Model-Based Cascade Ranking.

is better than that found by $\pi_2$. This illustrates a significant optimization objective gap between the RL-based formulation and the original true objective.

## 3.2 INCREMENTALLY LOCALIZED ACTIONS

**Non-smooth Objective Functions** We further empirically investigate the properties of the objective function $f$. As the domain of $f$ is the high-dimensional space $\mathcal{S}$, directly visualizing the landscape of the objective function is challenging. Instead, we sample a large number of diverse state points from the state space to approximate the function's behavior. Specifically, we visualize the training curve produced by a simple evolutionary algorithm (EA) that uses random actions from the RL action space $a \in \mathcal{A}$ to repeatedly perturb the current best solution locally (see Appendix E for details.) As shown in Figure 2 (Right), the sampled function values exhibit significant oscillations, indicating that the underlying objective function is highly oscillatory as well. The major reason for the oscillations of the evaluation function stems from the complex optimization mechanisms employed by circuit synthesis tools. Even minor modifications to the circuit structure can result in substantial performance variations when evaluated by these tools. Consequently, the oscillatory nature of the optimization objective results in numerous local optimal solutions.

**Limited Exploration Ability of Localized Actions** The oscillatory nature of the objective functions results in numerous local optimal solutions, thus requiring globally diverse exploration of the design space to escape local optima. However, the actions in the existing RL formulation are limited to local modifications of design structure at a certain column, severely constraining the agent's ability to explore diverse or distant regions of the search space. As a result, the search process may become confined to suboptimal regions, limiting the chances of discovering global optima.

## 4 A MODEL-BASED CIRCUIT GENETIC EVOLUTION FRAMEWORK

We start with an overview of our proposed MUTE in Section 4.1. Next, we outline the formal procedure at the core of MUTE, specifying the circuit genetic evolution formulation and population initialization in Section 4.2, our proposed efficient and effective circuit genetic variation operators in Section 4.3, and model-based cascade ranking for selection in Section 4.4.

### 4.1 OVERVIEW OF OUR FRAMEWORK

We provide an illustration of our MUTE in Figure 3. To bridge the gap between the RL-based formulation and the original task, we design a grid-based genetic representation of solutions, and reformulate the computing circuits optimization problem as a circuit genetic evolution process. The evolution formulation is an iterative process between circuit genetic variation and model-based selection, where each iteration evaluates and improves solutions using the true objective value, thus bridging the objective gap. First, we propose a learning-based population initialization method to accelerate the evolution process by leveraging the existing RL methods to generate a population of high-performing design solutions. Then, we propose efficient and effective genetic variation operators to avoid redundant exploration and promote globally diverse exploration. Finally, to further improve sample efficiency, we propose a model-based cascade ranking method for efficient selection from a large number of generated offspring design solutions.

## 4.2 Genetic Evolution Formulation and Population Initialization

**Circuit Genetic Evolution Formulation** For computing circuits such as adders and multipliers, we can represent each circuit design solution by a grid of numbers. We take a compressor tree design solution as an example. As the total number of different full and/or half adders at each column implicitly encodes the structure of a compressor tree, it significantly impacts the post-synthesis performance of multipliers (Xiao et al., 2021; Zuo et al., 2023). Specifically, we employ a $2 \times N_C$ grid to represent the compressor tree design solution, where two rows denote the total number of full and half adders across $N_C$ columns, respectively.

To enable circuit genetic evolution, we propose to formulate each element in the grid-based design solution representation as a circuit gene, and thus the grid as a genetic individual. Note that the existing RL-based method can formulate the grid-based design solution as the state representation as well. Thus, the grid-based genetic formulation allows us to seamlessly incorporate learning into our genetic evolution framework, which can significantly improve sample efficiency.

To directly optimize the original circuit design performance, we formulate the fitness function as the underlying design evaluation function $f$ given by circuit synthesis tools. Thus, the optimization objective of our circuit genetic formulation takes the form of $\arg\min_{s \in \mathcal{S}} f(s)$, where $\mathcal{S}$ denotes the set of all possible legal design solutions.

**Learning-Based Population Initialization** Although the performance of existing RL methods is limited by deceptive reward signals, they can efficiently converge to a local optimal design solution. Thus, to speed up the evolution process, we incorporate the learning method into our population initialization to generate a set of high-performing circuit design solutions. During the learning process, we maintain an *elite pool* of $N$ best-performing design solutions, and progressively update the pool at each training episode. Finally, we use the elite pool as an initial population.

Following previous work (Song et al., 2022; Zuo et al., 2023; Wang et al., 2024g), we use a scalarized version of the Deep Q-network (DQN) algorithm (Mnih et al., 2015) to learn solution modification policies. We maintain a progressively updated elite pool, and each episode starts the environment with a design solution $s_0$ sampled from the elite pool. Every action $a_t$ from the Q-network $Q_\theta$ modifies the design solution $s_t$ to another design solution $s_{t+1}$, and returns a weighted reward $r_t$ that indicates the decrease in the normalized circuit area and delay. That is,

$$r_t = w_a(\text{area}(s_t) - \text{area}(s_{t+1})) + w_d(\text{delay}(s_t) - \text{delay}(s_{t+1})). \tag{3}$$

After each episode, we insert each design solution at this episode into the elite pool when its design performance is better than the worst solution in the current elite pool.

## 4.3 Efficient and Effective Genetic Variation Operators

Designing efficient and effective genetic variation operators is important for the success of genetic algorithms (Zhu et al., 2023a; Bai et al., 2023b; Li et al., 2024a). Thus, we propose a sequential mutation operator for efficient long-term exploration via sequential modifications of design solutions based on the Q-network learned during initialization. We propose a multi-granularity crossover operator for globally diverse exploration by recombining two genetic solutions across diverse granularities. By using these genetic variation operators, we can generate a large set of offspring solutions from a population of design solutions at each iteration. We present details as follows.

**A Sequential Mutation Operator for Efficient Exploration** To explore the design solution space, a common idea is to perform random local modifications on design solutions. However, the random modification strategy can lead to redundant and myopic exploration, thus resulting in low sample efficiency and sub-optimal solutions. To promote efficient long-term exploration, we propose a sequential mutation operator to perform sequential local modifications on a given solution based on the learned Q-network. Specifically, for a design solution $s_0$ sampled from the current generation of populations, we leverage the learned Q-network $Q_\theta$ to guide episodic modifications on the design solution by sampling $T$ local modification actions for generating mutated solutions. Thus, we generate $T$ mutated offspring solutions by sampling a sequence of modification actions from the Q-network, i.e., $a_t = \arg\max_a Q_\theta(s_t, a), t = 0, \ldots, T$. To prevent premature convergence, we follow the $\epsilon$-greedy strategy (Sutton & Barto, 1998) to sample actions.

This sequential mutation operator offers two key advantages. (1) Guided by the learned Q-function, we can prioritize modifications that enhance design performance, rather than relying on random

modifications. (2) By leveraging the long-term predictive capabilities of the learned Q-function, we can identify strong combinations of local modifications, leading to more effective design solutions.

**A Multi-Granularity Crossover Operator for Diverse Exploration** As shown in Figure 2, the objective function $f$ is highly oscillatory, resulting in many local optimal design solutions. However, relying solely on existing local modifications severely restricts the exploration capability, as it only incrementally adjusts the number of full or half adders at a single column within a specific design solution. To enable global exploration for escaping local optima, it is crucial to develop design variation operators capable of making globally diverse exploration of the design space.

To this end, we propose a multi-granularity genetic crossover operator that recombines design substructures at varying column ranges between two grid-based genetic solutions. The key advantage of this approach is its ability to expand design variation from the single-gene level to the cross-column level. This allows for transitions from small changes involving a single column to more significant modifications that span multiple columns. This significantly enhances our global exploration capability, improving the ability to escape local optima.

*Single Gene/Column Crossover* Given two high-performing design solutions $s^0$ and $s^1$, we view the solutions as a sequence of genes, i.e., $s^0 = \left\{(g_{3:2}^0(0), g_{2:2}^0(0)), \cdots, (g_{3:2}^0(N), g_{2:2}^0(N))\right\}$ and $s^1 = \left\{(g_{3:2}^1(0), g_{2:2}^1(0)), \cdots, (g_{3:2}^1(N), g_{2:2}^1(N))\right\}$.

To recombine the two high-performing solutions to obtain diverse offspring solutions, the single column crossover randomly selects a column and recombines the genes at that column. Specifically, the column granularity crossover operator generates two children in the following form:

$$s_c^0 = \left\{\cdots, (\mathbf{g_{3:2}^1(i)}, \mathbf{g_{2:2}^1(i)}), \cdots\right\} \text{ and } s_c^1 = \left\{\cdots, (\mathbf{g_{3:2}^0(i)}, \mathbf{g_{2:2}^0(i)}), \cdots\right\}, \tag{4}$$

where $i$ is the selected column. Meanwhile, the single gene crossover only recombines the genes representing a certain adder type at the selected column.

*Cross Columns Crossover* Given two high-performing design solutions $s^0$ and $s^1$, the cross columns crossover randomly selects two columns and recombines the two grid-based genetic solutions within the selected two columns. Specifically, the cross columns crossover generates two offspring solutions in the following form:

$$s_c^0 = \left\{\cdots, (\mathbf{g_{3:2}^1(i)}, \mathbf{g_{2:2}^1(i)}), \cdots, (\mathbf{g_{3:2}^1(j)}, \mathbf{g_{2:2}^1(j)}), \cdots\right\} \text{ and}$$
$$s_c^1 = \left\{\cdots, (\mathbf{g_{3:2}^0(i)}, \mathbf{g_{2:2}^0(i)}), \cdots, (\mathbf{g_{3:2}^0(j)}, \mathbf{g_{2:2}^0(j)}), \cdots\right\}, \tag{5}$$

where $i$ and $j$ are the selected two columns. Depending on the values of $i$ and $j$, this crossover can recombine substructures across any different columns, thereby promoting global diverse exploration.

Note that the crossover operator could lead to illegal solutions. Thus, we design simple legalization rules following previous work (Zuo et al., 2023) to ensure that these generated children are legal design solutions. Please refer to Appendix F.5 for details.

### 4.4 MODEL-BASED CASCADE RANKING FOR EFFICIENT SELECTION

The success of genetic algorithms often relies on sampling a large number of solutions (Zhu et al., 2023a; Bai et al., 2023b; Li et al., 2024a). However, in circuit optimization tasks, evaluating each design solution using circuit synthesis tools is highly time-consuming, which significantly limits the number of samples for searching. To significantly improve sample efficiency, we propose learning a design evaluation model $\hat{f}_\Theta$ using the collected populations during evolution.

**Model Training** Inspired by standard model-based RL (Janner et al., 2019; Wang et al., 2023c), we first train a prediction model using the collected replay buffer during the population initialization process, and then adaptively update the model using a few populations with true evaluations during the evolution process. In terms of the model architecture, we employ the ResNet-18 as the state encoder and a multi-head decoder to predict the area and delay of the input state. The multi-head decoder comprises two multi-layer perceptrons (MLPs), each with two hidden layers with 256 units and ReLU activations. In terms of the training details, we use the mean squared error loss to update the model parameters. We use the Adam optimizer with a learning rate of 1e-3.

**Model Usage** Previous model-based RL methods (Janner et al., 2019; Wang et al., 2023c) have shown that directly using the learned model to replace the real environment suffers from model exploitation, i.e., overfitting to model errors. To address this challenge, we propose a model-based

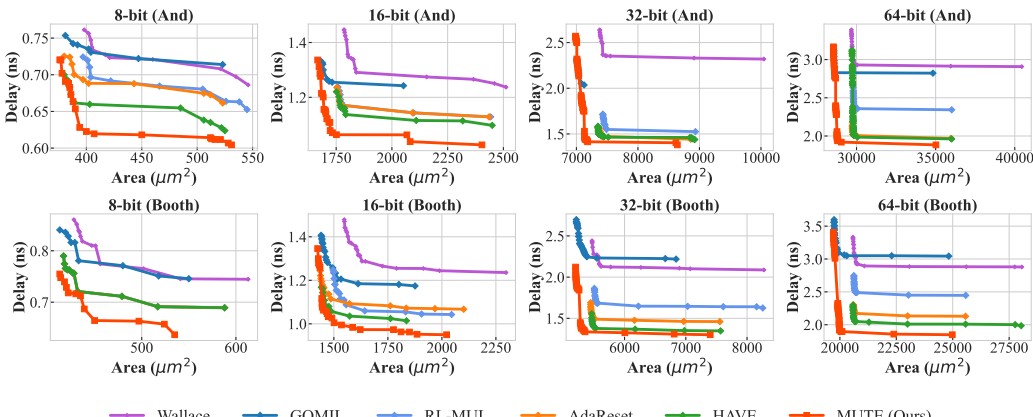

Figure 4: The results demonstrate that the multipliers optimized by MUTE consistently and significantly Pareto-dominate the designs optimized by all five baselines across eight multiplier design problems, improving the hypervolume by up to 38%.

cascade ranking method to efficiently and accurately select high-performing solutions from a large set of generated solutions. The key idea is to progressively select solutions through two-stage ranking models, where the first stage model is our learned model and the second stage model is the real circuit synthesis environment. Specifically, we primarily use the model to rapidly evaluate the children solutions generated by the designed genetic variation operators. We generate at least 100 children solutions at each iteration, and use the model to pre-rank these solutions. The top-5 solutions are then selected for evaluation in the true environment. This approach allows us to generate a substantial number of children solutions, promoting diverse global exploration.

## 5 EXPERIMENTS

We first introduce the experimental setup, baselines, and evaluation metrics in Section 5.1. Then, our experiments are designed with four primary objectives. 1) We evaluate the performance of MUTE in optimizing computing circuits, including multipliers, adders, and MACs, across a broad range of input widths in Section 5.2. 2) We investigate the generalization performance of multipliers optimized by MUTE to large-scale macros widely-used in real-world AI chips in Section 5.3. 3) We perform carefully designed ablation studies to demonstrate the importance of our genetic evolution formulation and provide insights into each component in MUTE in Section 5.4. 4) We conduct a thorough trade-off evaluation of the runtime and performance gains of MUTE in Section 5.5.

### 5.1 EXPERIMENTAL SETTINGS

**Experimental Setup** Throughout our experiments, we utilize the OpenROAD flow (Ajayi & Blaauw, 2019) alongside the NanGate 45nm open-cell library (Nangate Inc., 2008) for circuit synthesis, coupled with OpenSTA (Parallax Software Inc.) for timing analysis. The setting follows previous work (Zuo et al., 2023). These tools represent the state-of-the-art open-source EDA tools, and are widely used in research of EDA (Kahng, 2021; Tan et al., 2021; Pilipović et al., 2021). Our training procedure employs the Adam optimizer (Ruder, 2016) within the PyTorch framework (Paszke et al., 2019). For fair comparison, we controlled the training time of our method to be comparable with the baseline methods (see Appendix H.2). We apply our method to eight distinct multiplier design tasks, encompassing 8-bit, 16-bit, 32-bit, and 64-bit multipliers based on both AND gate-based and Booth encoding-based PPG techniques. Moreover, we apply our method to six distinct adder and MAC design tasks as well. We defer more results to Appendix H.

**Baselines** Our baselines encompass five competitive approaches, ranging from traditional human-designed heuristics to state-of-the-art (SOTA) learning-based methods. **1) Wallace Tree** (Wallace, 1964) represents a classical human-designed compression algorithm. **2) GOMIL** (Xiao et al., 2021) is an expert-designed algorithmic method based on integer programming. **3) RL-MUL** (Zuo et al., 2023), **4) AdaReset** (Song et al., 2022), and **5) HAVE** (Wang et al., 2024g) are three recent SOTA RL-based multiplier and adder optimization methods.

**Evaluation Metrics** The computing circuits optimization problem is an optimization task with multiple conflicting objectives, such as area and delay. Thus, we employ two widely-used multi-objective optimization evaluation metrics (Basaklar et al., 2022; Hung et al., 2023) to compare the

Table 1: The results demonstrate that MUTE significantly outperforms previous SOTA approaches in terms of the hypervolume on both adder and MAC design tasks.

| Adder | 16-bit | | 32-bit | | 64-bit | |
|---|---|---|---|---|---|---|
| Methods | HyperVolume ↑ | Improvement(%)↑ | HyperVolume ↑ | Improvement(%) ↑ | HyperVolume ↑ | Improvement(%) ↑ |
| RL-MUL | 88.03 | NA | 211.76 | NA | 503.45 | NA |
| AdaReset | 92.66 | 5.26 | 213.71 | 0.92 | 513.07 | 1.91 |
| HAVE | 113.92 | 29.41 | 254.09 | 19.99 | 566.26 | 12.48 |
| MUTE (Ours) | **116.94** | **32.84** | **281.66** | **33.01** | **582.73** | **15.75** |
| **MAC** | **16-bit** | | **32-bit** | | **64-bit** | |
| Methods | HyperVolume ↑ | Improvement(%)↑ | HyperVolume ↑ | Improvement(%) ↑ | HyperVolume ↑ | Improvement(%) ↑ |
| RL-MUL | 371.10 | NA | 4114.00 | NA | 6184.50 | NA |
| AdaReset | 369.60 | -0.40 | 5123.00 | 24.53 | 10212.62 | 65.13 |
| HAVE | 401.80 | 8.27 | 5221.24 | 26.91 | 10604.57 | 71.47 |
| MUTE (Ours) | **414.80** | **11.78** | **5843.00** | **42.03** | **11487.37** | **85.74** |

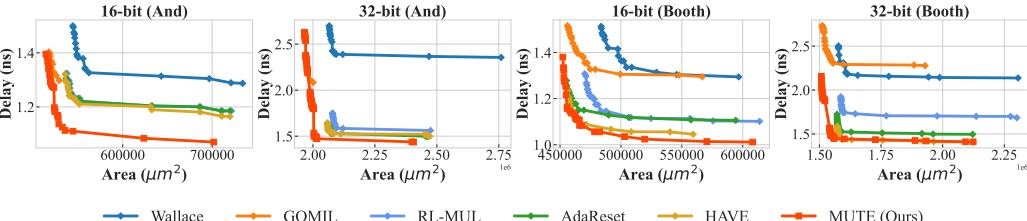

Figure 5: The results illustrate that PE arrays designed by MUTE consistently and significantly outperform the designs discovered by all five baselines in terms of Pareto-dominance across four multiplier design problems, i.e., 16-bit (And), 32-bit (And), 16-bit (Booth), 32-bit (Booth).

performance of our method with the baselines. First, we visualize the found Pareto points in terms of the area and delay for circuits designed by both our method and the baselines. Second, we utilize the hypervolume (HV) of the found Pareto points, which is defined by the volume of the region between a reference point and these found Pareto points. We defer more details to Appendix F.7.

## 5.2 Main Evaluation of Optimizing Computing Circuits

**Multiplier Design** We highlight the superiority of MUTE through a comparative analysis with five competitive baselines on eight multiplier design problems across a wide range of input sizes. The results in Figure 4 demonstrate that multipliers optimized by MUTE consistently and significantly outperform designs produced by all baselines across all eight multiplier design tasks. Moreover, we present the hypervolume of the Pareto points discovered by MUTE in Tables 6 and 7 in Appendix H.1. The results demonstrate that MUTE achieves a substantial improvement over the previous SOTA, improving the hypervolume by up to 38%. Overall, these results demonstrate the strong ability of MUTE to optimize multipliers, leading to significant reductions in both area and delay.

**Broad Applicability to Adders and MACs** To demonstrate that our approach is able to optimize a broad class of computing circuits, we apply our MUTE to optimizing two more fundamental computing circuits, i.e., adders and MACs. Specifically, we compare our MUTE with the three SOTA RL-based computing circuits optimization methods, i.e., RL-MUL (Zuo et al., 2023), AdaReset (Song et al., 2022), and HAVE (Wang et al., 2024g), on adders and MACs. As shown in Table 1, the results demonstrate that MUTE significantly outperforms previous SOTA approaches, improving the hypervolume by up to 42% compared to RL-MUL. The results not only highlight the superiority of our MUTE over previous SOTA approaches on optimizing computing circuits, but also demonstrate the broad applicability of our MUTE to a wide range of fundamental computing circuits.

## 5.3 Generalization to Large-Scale Circuits

To evaluate the generalization ability of our designed computing units to large-scale real-world computing circuits with numerous circuit units, we integrate these units optimized by MUTE and baselines into Processing Element (PE) arrays (Park & Chung, 2020; Son et al., 2023), which follows previous work (Zuo et al., 2023; Wang et al., 2024g). PE arrays are widely used in parallel computing tasks and large-scale data processing like Deep Neural Network (DNN) accelerators.

The results in Figure 5 show that PE arrays incorporating multipliers optimized by MUTE consistently and significantly Pareto-dominate those utilizing multipliers obtained from baselines. Furthermore, we present the hypervolume of the Pareto frontiers discovered by MUTE in Table 11 in Appendix H.4. The results demonstrate a significant improvement in hypervolume achieved by MUTE, outperforming previous SOTA by up to 48.18%. We provide detailed results in Appendix

Table 3: Analysis on the trade-off between performance gains and runtime of each module in MUTE.

| | 16-bit And | | | 32-bit And | | |
|---|---|---|---|---|---|---|
| Methods | HyperVolume | Improvement(%) | RunTime (h) | HyperVolume | Improvement(%) | RunTime (h) |
| Wallace | 332.91 | -25.96 | - | 1685.61 | -70.09 | - |
| HAVE | 505.83 | 12.50 | 20.33 | 5822.03 | 3.29 | 37.27 |
| SA | 449.63 | NA | 9.33 | 5636.51 | NA | 17.40 |
| CGE | 594.15 | 32.14 | 10.96 | 6285.00 | 11.51 | 18.35 |
| CGE+Learning | 608.25 | 35.28 | 15.17 | 6336.50 | 12.42 | 26.25 |
| CGE+Learning+Model (MUTE) | **622.55** | **38.46** | **17.33** | **6461.65** | **14.64** | **37.11** |

H.4. Overall, the results highlight the strong capability of MUTE to well generalize to large-scale computation-intensive circuits, thereby substantially improving real-world AI chips.

## 5.4 ABLATION STUDY

We present carefully designed ablation studies on multiplier design tasks as follows.

**Contribution of Each Component** To demonstrate the effectiveness of each component within MUTE, we conduct a thorough ablation study on multiplier design tasks.

In terms of the efficient and effective genetic variation module, we have designed two methods, called MUTE without Crossover (**w/o Crossover**) and MUTE without Mutation (**w/o Mutation**). MUTE without Crossover and MUTE without Mutation removes the designed genetic crossover and mutation operators, respectively. The results in Table 2 show that our designed genetic mutation and crossover operators are both critical for optimizing computing circuits, demonstrating the strong ability of the designed operators for promoting efficient and diverse exploration.

Table 2: The results demonstrate that each component within MUTE is significant.

| | 16-bit And | | 32-bit And | |
|---|---|---|---|---|
| Methods | HyperVolume ↑ | Improvement(%) ↑ | HyperVolume ↑ | Improvement(%) ↑ |
| Wallace | 332.91 | NA | 1685.61 | NA |
| MUTE (Ours) | **622.55** | **87.00** | **6461.65** | **283.34** |
| | Genetic Variation Module | | | |
| w/o Crossover | 585.30 | 75.81 | 5996.00 | 255.72 |
| w/o Mutation | 600.10 | 80.26 | 6385.00 | 278.79 |
| | Model-Based Module | | | |
| w/o Model | 605.20 | 81.79 | 6278.00 | 272.45 |
| | Learning Module | | | |
| w/o Learning | 578.10 | 73.65 | 5766.00 | 242.07 |

In terms of the model module, we have designed MUTE without Model (**w/o Model**) by removing the model-based module. The results in Table 2 show that learning a model can further improve the found designs in terms of the hypervolume. In terms of the learning module, we have designed MUTE without Learning (**w/o Learning**) by removing the learning module. The results demonstrate the significance of introducing learning into our genetic evolution for improving sampling efficiency.

## 5.5 PERFORMANCE GAIN AND RUNTIME TRADE-OFF EVALUATION

To further investigate the cost-effectiveness of each module of MUTE, we conducted an ablation study evaluating the trade-off between runtime and performance gains. MUTE consists of three modules: (1) the Circuit Genetic Evolution (CGE) module, (2) the Learning module, and (3) the Model-based module. The CGE module reformulates the multiplier design as a genetic evolution problem, using grid-based genetic representation and a random mutation policy. The Learning module introduces policy-guided population initialization and mutation. The Model-based module incorporates a learned model and a cascade ranking selection procedure.

Experimental results, summarized in Table 3, show that the CGE module significantly improves performance, achieving up to 32.14% gain with minimal runtime increase. The Learning module, while increasing runtime due to its computational overhead, enhances exploration efficiency and improves hypervolume. The Model-based module further improves hypervolume by 3.18%, though it increases runtime due to model training and additional sample collection. Despite these increases, the total runtime of MUTE remains comparable to the recent state-of-the-art approach, HAVE, highlighting the balance between performance gains and computational costs.

## 6 CONCLUSION

In this paper, we theoretically and empirically show a significant objective gap between the existing RL-based formulation and the original task due to the deceptive reward signals and incrementally localized actions in the RL-based formulation. To address this challenge, we propose a novel **m**odel-based circ**u**it gene**t**ic **e**volution (MUTE) framework, which reformulates the problem as a genetic evolution process by proposing a grid-based genetic representation of design solutions. Experiments on these circuits demonstrate that MUTE significantly Pareto-dominates state-of-the-art approaches in terms of both area and delay, improving the hypervolume by up to 38%.

# 7 ACKNOWLEDGEMENTS

The authors would like to thank all the anonymous reviewers for their insightful comments and valuable suggestions. This work was supported by the National Key R&D Program of China under contract 2022ZD0119801 and the National Nature Science Foundations of China grants U23A20388, 62021001 and 624B1011.

## REPRODUCIBILITY STATEMENT

In this study, to ensure the reproducibility of our approach, we provide key information from the main text and Appendix as follows.

1. **Algorithm.** We provide the architecture and illustration of our MUTE in Figure 3 and Section 4. We also provide the detailed implementation of MUTE in Appendix F. See Appendix F.6 for the hyperparameters of MUTE.
2. **Source Code.** To facilitate the evaluation process and support a thorough review, we have released our source code at the following link: https://anonymous.4open.science/r/AI4MUL-4199.
3. **Experimental Details.** We provide detailed experiment settings in Section 5.1.
4. **Theoretical Proofs.** We provide all proofs in Appendix A.

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

## A  THEORETICAL ANALYSIS

This section presents a theoretical analysis of the significant misalignment between the result of the optimal policy obtained through RL and the original optimal circuit design. First, the proof of Theorem 3.1 is provided in subsection A.1. Second, subsection A.2 offers further theoretical insights into the deceptive nature of the existing reward formulation.

Throughout the following theoretical analysis, we assume that the state space $\mathcal{S}$ is finite, and there is a terminal action for each state that can terminate the episode at this state. The assumptions usually hold in practical multiplier optimization problems. We focus on deterministic policies.

### A.1  PROOF OF THEOREM 3.1

**Lemma A.1.** *For any policy $\pi$ in the MDP, there exists a policy $\pi'$ that terminates at a certain state such that $R^{\pi'} \geq R^{\pi}$.*

*Proof.* Since state space $\mathcal{S}$ is finite, the states in trajectory generated by policy $\pi$ are finite as well. Therefore, there exists a state $s_T = \arg\min_{s \in \tau_{\pi}} f(s)$, where $\tau_{\pi}$ is the set of states in the trajectory of $\pi$ and $T$ is a finite number. We denote the trajectory $\tau_{\pi}$ by $\{s_0, s_1, \ldots, s_T, \ldots\}$.

Then we construct a new policy $\pi'$ that generates a trajectory $\tau_{\pi'} = \{s'_0, \ldots, s'_T\}$, where $s'_t = s_t, \forall t \leq T$. Note that the trajectory $\tau_{\pi'}$ terminates at the state $s_T$. Then we have

$$R^{\pi'} = \sum_{t=0}^{T-1} \gamma^t (f(s_t) - f(s_{t+1})) \tag{6}$$

$$= \sum_{t=0}^{T-1} \gamma^t f(s_t) - \sum_{t=0}^{T-1} \gamma^t f(s_{t+1}) \tag{7}$$

$$= f(s_0) - \sum_{t=0}^{T-2} (\gamma^t - \gamma^{t+1}) f(s_{t+1}) - \gamma^{T-1} f(s_T) \tag{8}$$

$$= f(s_0) - (1-\gamma) \sum_{t=0}^{T-2} \gamma^t f(s_{t+1}) - (1-\gamma) \sum_{t=T-1}^{\infty} \gamma^t f(s_T) \tag{9}$$

$$\geq f(s_0) - (1-\gamma) \sum_{t=0}^{T-2} \gamma^t f(s_{t+1}) - (1-\gamma) \sum_{t=T-1}^{\infty} \gamma^t f(s_{t+1}) \tag{10}$$

$$= R^{\pi} \tag{11}$$

$\square$

Then we prove Theorem 3.1 as follows.

*Proof.* Recall that Theorem 3.1 states that "The optimal RL policy $\pi^* := \arg\max_{\pi} R^{\pi}$ terminates at a state, and the state is a local optimal state of the evaluation function $f$." We prove this Theorem in the following two steps.

(1) $\pi^*$ **has a terminal state** If the optimal policy $\pi^*$ doesn't terminate, then according to Lemma A.1 there exists a distinct policy with a higher return, which contradicts with the definition of the optimal policy $\pi^*$. Thus, $\pi^*$ terminates at a certain state.

(2) **Local Minimality** Denote the terminate state of $\pi^*$ as $s_T$. By contradiction, suppose $\exists a_0 \in \mathcal{A}, a_0 \neq$ terminate such that $f(T(s_T, a_0)) < f(s_T)$. Then consider a new policy $\pi'$ whose trajectory is identical to $\pi$ before the $T$-th step but execute action $a_0$ rather than terminating at $T$-th step, and

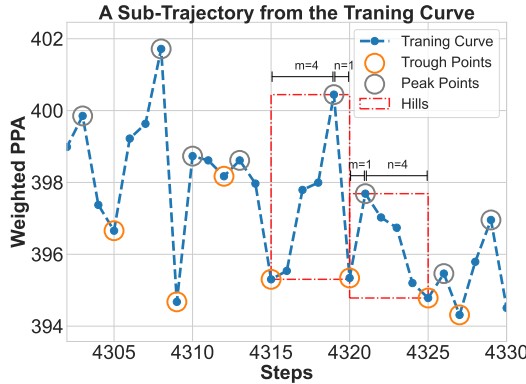

Figure 6: Sampled points from the training curve illustrating the oscillatory properties of the evaluation function.

execute terminate at step $T + 1$. Then the return of policy $\pi'$ is

$$R^{\pi'} = \sum_{t=0}^{T} \gamma^t (f(s_t) - f(s_{t+1})) \tag{12}$$

$$= R^{\pi^*} + \gamma^T (f(s_T) - f(T(s_T, a_0))) \tag{13}$$

$$> R^{\pi^*} \tag{14}$$

which contradicts the definition of the optimal $\pi^*$. □

### A.2 MULTIPLE HIGH HILLS CONDITION FOR GAP EXISTENCE

Theorem 3.1 indicates that the existing reward formulation guides the RL agent to evolve the circuit from an initial design to a locally optimal solution. However, the conditions under which this local optimum is the same as the global optimum remain unclear. In this subsection, we provide heuristic conclusions based on the oscillatory behavior of the objective function.

First, we sample a trajectory from the sigh-dimensional solution space using a basic EA algorithm for the purpose of visualization and simplified analysis, denoted as $\tau_{Sim}$, as shown in Figure 2. Our analysis focuses on the impact of the proxy reward function on the optimization objective along the sampled one-dimensional function curve, avoiding the complexities of the high-dimensional state space. Using the sampled trajectory $\tau_{Sim}$, we define a simplified MDP (Sim-MDP) with the tuple $(\mathcal{S}_{Sim}, \mathcal{A}_{Sim}, T_{Sim}, r, \gamma, \mu_0)$. Here, $\mathcal{S}_{Sim} = \tau_{Sim}$. The action space is simplified into $\mathcal{A}_{Sim} = \{Go, Terminate\}$. The transition function is defined as $T_{Sim}(s_t, Go) = s_{t+1}$, and the episode terminates upon executing the terminate action. The reward function, discount factor, and initial state are consistent with the original MDP. Any policy in this Sim-MDP corresponds to a trajectory terminating at a certain state. The Sim-MDP simplifies state and action spaces while keeping the reward function unchanged, facilitating the analysis of the optimization objective gap.

Based on the Sim-MDP, we calculate the cumulative discounted rewards of policies starting from the initial state and terminating at various points along the curve. We then compare the returns of these policies with the return of the policy $\pi_{s^*}$ that converges to the global minimum point $s^*$ in the Sim-MDP. For discount factors of 0.8, 0.9, and 0.99, the proportions of policies with returns higher than $\pi_{s^*}$ accounts for approximately 54.14%, 54.07%, and 45.71%, respectively. Moreover, for discount factors of 0.8, 0.9, and 0.99, the relative objective gap between the optimal RL policy and the global optimal point are roughly 35.52%, 34.75%, and 26.19%, respectively. This demonstrates a significant gap between the RL proxy optimization objective and the original optimization objective, attributed to the highly oscillatory nature of the optimization objective function.

Second, to further understand the reasons for this gap, we theoretically show conditions for the existence of optimization objective gap in the Sim-MDP, based on the observation that the evaluation function is highly oscillatory, leading to many peak and trough points. We first rigorously formulate the evaluation function $f$ across the sampled trajectory $\tau_{Sim}$ by using many repeated hills, which is inspired by the properties of oscillatory functions. A state $s_t \in \mathcal{S}_{Sim}$ is a **Peak Point** if $f(s_t) >$

$f(s_{t-1})$ and $f(s_t) > f(s_{t+1})$. In contrast, a state $s_t \in \mathcal{S}_{\text{Sim}}$ is a **Trough Point** if $f(s_t) < f(s_{t-1})$ and $f(s_t) < f(s_{t+1})$. Without loss of generality, we assume the initial state of Sim-MDP is a through point. We denote $p_i$ as the step index of the $i$-th peak point, and $t_i$ as the step index of the $i$-th through point. The $i$-th **Hill** is defined by a set of states between the $i$-th trough point and the $(i+1)$-th trough point, i.e., $P_i := \{s_t | t_i \leq t \leq t_{i+1}\}$. We denote the number of ascending steps $p_i - t_i$ by $m_i$, and the number of descending steps $t_{i+1} - p_i$ by $n_i$. Inspired by the fact that the action will always affine the structure of the multiplier which changes the evaluation function, for the $i$-th Hill we denote the lower bound of the variation of $f$ as $\delta_i := \min_{j \in [t_i, p_i)} \{|f(s_{j+1}) - f(s_j)|\}$; on the other hand the evaluation function $f$ is bounded, thus we denote the upper bound of the variation of as $\epsilon_i := \max_{j \in [p_i, t_{i+1})} \{|f(s_j) - f(s_{j+1})|\}$. The formulation is illustrated in Figure 6. With the multiple hills formulation, we provide a Multiple High Hills Condition for the existence of optimization objective gap.

**Theorem A.2** (Multiple High Hills Condition). *Denote the number of hills in the Sim-MDP before $s^*$ as $N$, where $s^* := \arg\min_{s \in \mathcal{S}_{sim}} f(s)$ is the global optimal state in Sim-MDP. If there exists the $i$-th trough point $s_{t_i}$ such that*

$$\sum_{j=i}^{N} \gamma^{t_j - t_i} \left[ \frac{\gamma^{m_j}(1 - \gamma^{n_j})}{1 - \gamma} \epsilon_j - \frac{1 - \gamma^{m_j}}{1 - \gamma} \delta_j \right] < 0, \tag{15}$$

*then the optimization objective gap exists, i.e., the optimal policy in Sim-MDP $\pi^*_{Sim}$ converges to a sub-optimal solution rather than the global optimal state $s^*$.*

*Proof.* Suppose $s^*$ is achieved at step $T$, and denote the policy that terminates at $s^*$ as $\pi_0$. For any through point $s_{t_i}$, consider a new policy $\pi'$ that terminates at $s_{t_i}$ and its return is

$$R^{\pi'} = \sum_{t=0}^{t_i - 1} \gamma^t (f(s_t) - f(s_{t+1})) \tag{16}$$

Then we have

$$R^{\pi_0} - R^{\pi'} = \sum_{t=t_i}^{T-1} \gamma^t (f(s_t) - f(s_{t+1})) \tag{17}$$

$$= \sum_{j=i}^{N} \sum_{t=t_j}^{t_{j+1}-1} \gamma^t (f(s_t) - f(s_{t+1})) \tag{18}$$

$$= \sum_{j=i}^{N} \left\{ \sum_{t=t_j}^{t_j + m_j - 1} \gamma^t (f(s_t) - f(s_{t+1})) + \sum_{t=t_j + m_j}^{t_j + m_j + n_j - 1} \gamma^t (f(s_t) - f(s_{t+1})) \right\} \tag{19}$$

$$= \sum_{j=i}^{N} \left\{ \gamma^{t_j} \sum_{t=0}^{m_j - 1} \gamma^t (f(s_t) - f(s_{t+1})) + \gamma^{t_j + m_j} \sum_{t=0}^{n_j - 1} \gamma^t (f(s_t) - f(s_{t+1})) \right\} \tag{20}$$

$$\leq \sum_{j=i}^{N} \left\{ \gamma^{t_j} \sum_{t=0}^{m_j - 1} \gamma^t (-\delta_i) + \gamma^{t_j + m_j} \sum_{t=0}^{n_j - 1} \gamma^t \epsilon_i \right\} \tag{21}$$

$$= \sum_{j=i}^{N} \gamma^{t_j} \left\{ \frac{1 - \gamma^{m_j}}{1 - \gamma} (-\delta_i) + \gamma^{m_j} \frac{1 - \gamma^{n_j}}{1 - \gamma} \epsilon_i \right\} \tag{22}$$

$$\tag{23}$$

Then $\sum_{j=i}^{N} \gamma^{t_j} \left\{ \frac{1 - \gamma^{m_j}}{1 - \gamma} (-\delta_i) + \gamma^{m_j} \frac{1 - \gamma^{n_j}}{1 - \gamma} \epsilon_i \right\} < 0 \Rightarrow R^{\pi_0} < R^{\pi'}$, i.e. $\pi'$ has higher return, and further the optimal policy in Sim-MDP $\pi^*_{\text{Sim}}$ converges to a sub-optimal solution rather than the global optimal state $s^*$. $\square$

This Theorem implies that if there are multiple high hills, i.e., unexpected high peak points (large $\delta_j$), on the trajectory between a trough point (i.e., a local optimum) and the global optimum, then

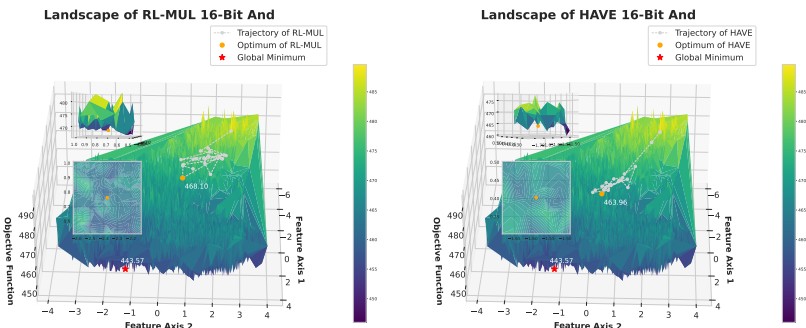

Figure 7: Optimization objective landscape of 16-bit and gate-based multiplier. The global minimum is marked as the red point. The grey lines mark RL-MUL and HAVE trajectories, with points indicating the optimal solutions found every five episodes. The optimum found by RL-MUL and HAVE is marked as the yellow point, with the objective value annotated nearby.

the RL optimization objective converges to the local optimum rather than the global optimum, due to the noisy information in the cumulative discounted performance objective from those unexpected high peak points.

## B  Visualization of Optimization Objective Landscape

In this section, we visualize the optimization objective landscape using the 16-bit AND gate-based multiplier optimization task.

**Visualizing optimization objective landscape**

(1) **Data Collection** To ensure comprehensive sampling and maximize coverage of the solution space, we first employ the RL-MUL (Zuo et al., 2023) and HAVE (Wang et al., 2024g) algorithms to generate initial populations, each comprising 2,500 solutions. Building on this initial population, we extensively apply our genetic variation operators, ultimately producing a total of 50,000 solutions.

(2) **Data Visualization** To visualize the high-dimensional solution space, we use Principal Component Analysis (PCA) (Abdi & Williams, 2010) to reduce its dimensionality to two dimensions. This enables the creation of a 3D surface plot depicting the relationship between solutions and their fitness values, i.e., our optimization objective landscape. We reconstruct the objective function surface from the data points using Delaunay triangulation, which is widely used in surface reconstruction from a set of points (Amenta et al., 1998; 2000; Cazals & Giesen, 2006).

(3) **Results** As shown in Figure 7, the results reveal that the optimization surface is highly oscillatory and characterized by numerous local optima. Using the visualization method described above, we transformed the points collected during the training processes of RL-MUL and HAVE into curves. The results reveal that both RL-MUL and HAVE converged to specific local optima.

**The convergence into local optima of RL-MUL and HAVE**

Using the visualization method described above, we transformed the points collected during the training processes of RL-MUL and HAVE into curves. The results reveal that both RL-MUL and HAVE converged to specific local optima.

## C  Related Work

### C.1  Computing Circuits Optimization

Computing circuits like adders and multipliers are widely employed in practical applications, leveraging Compressor Tree and Prefix Tree for efficient parallel operations. Generally, optimization methods for these circuits can be categorized into three main approaches. (1) **Manual designs** involve leveraging human expertise to craft architectures derived from regular designs, which require

substantial engineering effort. Various compressor Trees have been devised to reduce partial products, as shown in (Wallace, 1964; Dadda, 1983; Itoh et al., 2005; Oklobdzija et al., 1996), while Prefix Trees are optimized for more efficient parallel addition, as demonstrated in (Beaumont-Smith & Lim, 2001; Sklansky, 1960; Brent & Kung, 1982). (2) **Conventional algorithmic** methods (Xiao et al., 2021; Zuo et al., 2024a; Liu et al., 2003; Roy et al., 2013) generate circuit architectures using specific strategies such as mathematical programming and heuristic search. However, they often optimize circuits using proxy metrics such as size and depth, which may result in a significant discrepancy from actual performance in the real design flow. (3) Recent methods (Zuo et al., 2023; 2024b; Roy et al., 2021; Song et al., 2022; Wang et al., 2024g; Lai et al., 2024) propose using **reinforcement learning** to optimize circuits based on the post-synthesis metrics, incorporating synthesis into the optimization loop, This offers promising approaches to bridge the gap between proxy metrics and actual performance. In this paper, we focus on optimizing computing circuits using post-synthesis metrics as well.

## C.2 REINFORCEMENT LEARNING

Reinforcement Learning (RL) has achieved great success in sequential decision-making problems, encompassing applications from video game playing to robotic control(Mnih et al., 2015; Kaiser et al., 2020; Duan et al., 2016; Zhang et al., 2024; Liu et al.). RL approaches can be generally divided into model-free Haarnoja et al. (2018); Wang et al. (2023e); Yang et al. (2022b); Liu et al. (2024d); Wang et al. (2023b); Liu et al. (2021); Yang et al. (2022a); Liu et al. (2023d), model-based Janner et al. (2019); Liu et al. (2023f); Wang et al. (2022), and offline RL Hu et al. (2021); Chen et al. (2024b); Jia et al. (2024); Liu et al. (2023e); Yang et al. (2024); Liu et al. (2024c) approaches. In this paper, our MUTE falls into the model-based category.

## C.3 GENETIC EVOLUTION ALGORITHMS

Genetic evolutionary algorithms (GA) are one of the most established and famous optimization methods, encompassing a diverse range of variants that find extensive applications across various fields (Garai, 2022; Alhijawi & Awajan, 2024). Inspired by Darwinian theories of species evolution in nature, genetic algorithms utilize selection, crossover, and mutation operators to evolve solutions, ultimately achieving global optimization (Slowik & Kwasnicka, 2020). Recently, Evolutionary Reinforcement Learning algorithms (ERLs) have emerged as a promising solution, effectively integrating the strengths of both reinforcement learning and evolutionary algorithms (Zhu et al., 2023a; Bai et al., 2023b; Li et al., 2024a). In this paper, we propose a learning-based population initialization, a sequential mutation operator, a multi-granularity crossover operator, and a model-based cascade ranking within a genetic algorithm framework for optimizing computing circuits.

## C.4 MACHINE LEARNING FOR CHIP DESIGN

With the exponential growth in chip complexity driven by advances in semiconductor technology, the application of machine learning (ML) to assist in the automated chip design workflow has garnered significant attention in recent years (Mirhoseini et al., 2021; Huang et al., 2021; Sánchez et al., 2023; Neto et al., 2021; Lai et al., 2022; 2023). The chip design process encompasses several stages (Huang et al., 2021; Ren & Hu, 2023), including high-level synthesis (Yao et al., 2024a; Liu et al., 2022; Yao et al., 2024b), logic synthesis (Li et al., 2023; Zhu et al., 2023b; Li et al., 2024c; Liu et al., 2023b;c; 2024b; Bai et al., 2025; Wang et al., 2024f;d; Chen et al., 2024a), placement (Lai et al., 2022; 2023; Geng et al., 2024; Chen et al., 2023; Shi et al., 2023a; 2025a; Wang et al., 2024e; Geng et al., 2025b), and design space exploration (Chen et al., 2024c; Bai et al., 2023a; 2021), among others.

## C.5 MACHINE LEARNING FOR COMBINATORIAL OPTIMIZATION

Optimizing multiplier circuit designs is also essentially a combinatorial optimization problem. The use of machine learning to tackle combinatorial optimization problems has been an active topic of significant interest in recent years (Bengio et al., 2021; Gasse et al., 2019; Shi et al., 2025b; 2023b; 2024; Geng et al., 2023; Wang et al., 2023d; 2024b; Ling et al., 2024; Li et al., 2024d; Wang et al.,

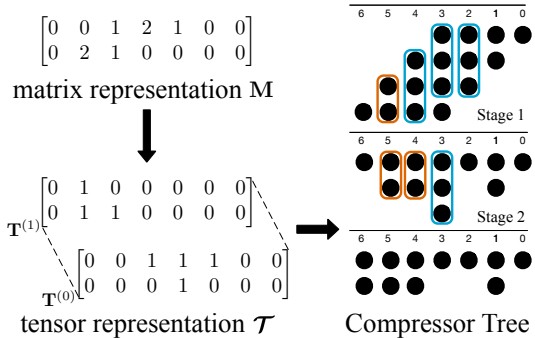

Figure 8: State representation in RL-MUL.

2021; 2024a; 2023a; Wang & Yu, 2023; Wang et al., 2024c; Geng et al., 2025a; Liu et al., 2023a; 2025; 2024a; Li et al., 2024f;e).

# D  IMPLEMENTATION DETAILS OF RL-MUL

**State Representation** We use the total number of 3:2 and 2:2 compressors in each column to present the multiplier structure. As illustrated in Figure 8, 4-bit multiplier structure and its matrix representation $M$ are shown. Given a matrix $M \in \mathbf{R}^{K \times (N_b + M_b)}$, where $m_{ij}$ indicates the quantity of the $i$-th compressor used in column $j$. RL-MUL follows a fixed scheme to extend the matrix $M$ to a tensor to obtain a unique representation for the assignment of compressors in multiplier stages. As shown in Figure 8, RL-MUL utilizes $\mathcal{T} \in \mathbf{R}^{K \times (M_b + N_b) \times ST}$ to represent a multi-stage state, where $K$ is the total types of compressors, $ST$ is the number of compression stages, and $N_b$ and $M_b$ is the input width. For any element $t_{ij}^k$ within $\mathcal{T}$, it signifies the utilization of the $i$-th type of compressor at column $j$ and stage $i$. The assignment method is to assign the compressors from the least significant bit (LSB) columns to the most significant bit (MSB) columns and assign the 3:2 compressors first as many as possible. After assigning the 3:2 compressors, if at column j there are still more than two PPs, it assigns the 2:2 compressors. Repeat this progress until all compressors are assigned. For example at column 4 in Figure 8, we first assign a 3:2 compressor in the first stage, then assign a 2:2 compressor in the second stage.

**Legalization Rules** When selecting actions, RL-MUL exclusively considers whether the action reduces the final production products to either 1 or 2. RL-MUL has four actions, including adding a 3:2 compressor, removing a 2:2 compressor, replacing a 3:2 compressor, and replacing a 2:2 compressor. Furthermore, an action performed at column j will have an impact on column j+1 and cause column j+1 illegal due to the propagation of the carry bit. RL-MUL employs a legalization strategy that refines the state from column j+1 to the most significant bit, ensuring that the PPs in every line are reduced to 1 or 2 following the actions. The strategy adds a 3:2 or replaces a 2:2 compressor if there is an over of PPs, and deletes a compressor if there is a lack of PPs.

# E  IMPLEMENTATION DETAILS OF THE BASELINES

**GOMIL** (Xiao et al., 2021) is a global optimization method that simultaneously considers the CT and CPA. The author provides the open-source C++ code. We can extract the required structure from its solution files.

## E.1  RL-BASED BASELINES

**RL-MUL** (Zuo et al., 2023) encodes the state into a tensor $\mathcal{T}$ described in Appendix D, using ResNet-18 as the network backbone and training based on the DQN algorithm. Different from the Random method, RL-MUL only chooses the action randomly in warm-up steps. In future steps, it chooses the action that can maximize the masked Q-value of the network.

**MBPO** Janner et al. (2019) is a state-of-the-art model-based RL method, which can significantly improve sample efficiency by learning an environment model. We implement the algorithm in our multiplier optimization environment by setting the Update-To-Data ratio as five.

## E.2 Evolutionary Algorithms

**MFEA** Slowik & Kwasnicka (2020) is a population-based global optimization method inspired by biological evolution. We maintain a population of candidate solutions, iteratively evolving them through random mutations, crossovers, and selections. The population is then updated based on the fitness values of the individuals.

**MBBO** Garnett (2023) is a global optimization method. We model the solution space as a high-dimensional vector space. We first sample a trajectory using a random walk. Then during each iteration, we fit an RBF kernel Gaussian Process model and use UCB (Upper Confidence Bound) as the acquisition function to determine the next sampling point.

## F Implementation Details of Our MUTE

### F.1 Hardware Specification

Our experiments were executed on a Linux-based system equipped with a 3.60 GHz Intel Xeon Gold 6246R CPU and NVIDIA RTX 3090 GPU.

### F.2 Synthesis Tool Setup

Nangate45 is a widely used standard cell library in the semiconductor industry. It is open source and free, and we can obtain it at `https://silvaco.com/services/library-design/`. Readers can refer to `https://github.com/The-OpenROAD-Project/OpenROAD-flow-scripts`, seeking the artifact of OpenROAD flow matched with the distribution.

In terms of the verilog generation, previous work uses EasyMAC Zhang et al. (2022) to implement it. We encode our CT following EasyMAC Zhang et al. (2022) rules which use a sequence $s_{ct} = p_0 p_1 \cdots p_r$ to represent a CT. Each $p_i = (index_i, type_i)$ signifies the index and type of a compressor. Considering that generating Verilog HDL codes by EasyMAC and running the logical synthesis are still time-consuming, we directly generate multiplier Verilog codes using our designed template. Compared to EasyMAC, our method can generate verilog code faster. To ensure fairness in comparison, we have employed a uniform default adder to implement CPA for all methods.

### F.3 Details on the Learning-Based Population Initialization

#### F.3.1 Best-Case Learning

In terms of population Initialization, we propose a best-case learning module, which maintains an elite pool with 20 currently found best design solutions for enhanced diversity. We restart the initial state by sampling a state from the elite pool at the beginning of each episode. In terms of the RL algorithms, we follow previous work (Zuo et al., 2023; Wang et al., 2024g) to use a DQN agent to learn Q-functions for selecting modification actions. In terms of the Q-network architecture, we use the ResNet-18 as the tensor state encoder, and use a multi-layer perceptron (MLP) to predict Q-values for each candidate action. The MLP contains two hidden layers with 256 units and the ReLU activation function. To train the Q-network, we use an Adam optimizer, and set the learning rate as 1e-4. For a fair comparison, we set hyperparameters to align with previous work (Zuo et al., 2023; Wang et al., 2024g).

#### F.3.2 Evaluation Model Learning and Conservative Model Usage

In terms of the model architecture, we employ the ResNet-18 as the tensor state encoder and a multi-head decoder to predict the area and delay of the input state. The multi-head decoder comprises two multi-layer perceptrons (MLPs), each with two hidden layers with 256 units and ReLU activation.

In terms of the training details, we use the mean squared error loss to update the model parameters. We use the Adam optimizer with a learning rate of 1e-3.

In terms of model usage, we primarily use the model to fast evaluate the children solutions generated by the designed genetic variation operators. Specifically, we generate at least 100 children solutions at each iteration, and use the model to pre-rank these solutions. The top-5 solutions are then selected for evaluation in the true environment. This approach allows us to generate a substantial number of children solutions, promoting extensive globally diverse exploration. Note that we do not use the model in the RL learning, as it will suffer from the **cumulative multi-step model errors** due to the sequential characteristics of RL methods. In contrast, using the model in our evolution process only suffers from **single-step model errors**.

**RunTime of Design Evaluation** As demonstrated in Table 4, our learned model can significantly reduce the design evaluation time compared to calling synthesis tools. RL-MUL (Zuo et al., 2023) employs EasyMAC and OpenRoad, while Vgen refers to a Verilog generation method we implemented, detailed in Appendix F.2. Al-

Table 4: Runtime comparison

|  | RunTime (s, every 100 samples) | |
| --- | --- | --- |
| Method/Circuit | 32-bit And | 64-bit And |
| EasyMAC+OpenRoad | 2930 | 10930 |
| Vgen+OpenRoad | 303 | 973 |
| Model (Ours) | **1.4** | **1.44** |

though Vgen considerably accelerates evaluation compared to EasyMAC, it remains inefficient when evaluating large volumes of design solutions.

### F.4 DETAILS ON THE RL-GUIDED MUTATION OPERATOR

**The action space** The actions consist of four types of local modifications to a Compressor Tree solution at a specific column. These modifications include adding a 2:2 compressor, removing a 2:2 compressor, replacing a 3:2 compressor with a 2:2 compressor, and replacing a 2:2 compressor with a 3:2 compressor.

**Q-network model** Our Q-network comprises a ResNet-18 (He et al., 2016) as an encoder to represent the input state, and a multi-layer perceptron as a decoder to predict Q-values. The input state is a grid-based genetic representation of the design solution. The output comprises the state-action values for each action.

**The learning process** We employ the Deep Q-network (DQN) algorithm (Mnih et al., 2015) to train the Q-network. During the population initialization phase, we train the Q-network using collected interactions with the circuit synthesis environment. We then periodically update the Q-network by sampling some interactions with the circuit synthesis environment throughout the evolution process.

**Managing invalid designs** We indeed apply a legalization rule to transforming any invalid design solution into a valid solution, which is designed by RL-MUL (Zuo et al., 2023). Specifically, a valid design requires each column to have exactly one or two remaining partial products after compression. Invalid designs—resulting from actions that impact subsequent columns—occur when a column has either zero or three remaining partial products. To resolve this, we implement a legalization process that starts from the affected column and progresses toward the most significant column. For columns with three remaining partial products, a 2:2 compressor is either replaced with or augmented by a 3:2 compressor. For columns with zero remaining partial products, a 2:2 or 3:2 compressor is removed, as appropriate. This process ensures that all columns maintain a valid number of remaining partial products (either one or two).

### F.5 DETAILS ON THE MULTI-GRANULARITY CROSSOVER OPERATOR

**Legalization Mechanism** Note that the crossover operators may lead to illegal solutions. Thus, we need to design a reasonable and simple legalization mechanism. Fortunately, we can follow the legalization rule proposed in RL-MUL (Zuo et al., 2023) to legalize these illegal solutions. Specifically, given a state that is modified from column i to any column, we can refine the state from column i to the most significant bit, ensuring that the final partial products (PPs) in every line are reduced to 1 or 2 following the actions. The legalization rule adds a 3:2 or replaces a 2:2 compressor if there is an over of PPs, and deletes a compressor if there is a lack of PPs.

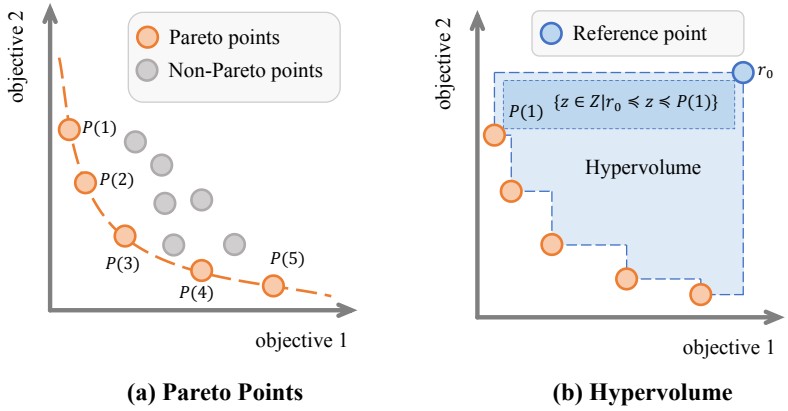

(a) Pareto Points                 (b) Hypervolume

Figure 9: **(a)** An example for a Pareto optimal set with 2 objectives and 5 Pareto optimal solutions (Pareto points). **(b)** An example for hypervolume with a selected reference point $r_0$. Integrated area $H(P, r_0)$ is the union of the rectangular areas where the reference point $r_0$ and the Pareto point $P(i)$ are diagonally opposite corners.

### F.6 ADDITIONAL COMMON HYPERPARAMETERS

In the above sections, we have provided implementation details and hyperparameters. Here, we list the common parameters used in the comparative evaluation and ablation study in Table 5. Note that we use the same hyperparameter as that of previous work (Zuo et al., 2023; Wang et al., 2024g) if possible for fair comparison.

Table 5: Common parameters used in the comparative evaluation and ablation study.

| Parameter | Value |
|---|---|
| Learning-Based Population Initialization Module | |
| environment steps per learning episode | 25 |
| policy updates per environment step | 1 |
| optimizer | Adam |
| discount ($\gamma$) | 0.8 |
| total learning episodes for initialization | 40 |
| Genetic Variation Module | |
| samples generated by sequential mutation operator at each iteration | 100 |
| samples generated by genetic crossover operator at each iteration | 200 |
| total iterations for evolution | 400 |
| Model-Based Module | |
| samples for circuit synthesis evaluation at each iteration | 5 |

### F.7 EVALUATION METRICS

Indeed, the multiplier optimization problem is a multi-objective optimization task with multiple conflicting objectives, such as area and delay. Thus, we use two evaluation metrics to compare our method with baselines. First, we visualize the approximated Pareto front in terms of the area and delay for multipliers designed by our method and baselines. Second, we use the hypervolume of the approximated Pareto front. We present details on the two metrics as follows.

**Multi-Objective Optimization Metrics**

Without loss of generality, considering a maximization optimization problem in n objectives, we aim to find the set of optimal solutions known as the Pareto optimal set. For an $n$-objective optimization problem, a solution $x$ Pareto dominates another solution $y$ if $x$ is not worse than y in all objectives and has at least one strictly better value, i.e., $\forall i \in [1, n], f_i(x) \geq f_i(y) \wedge \exists i \in [1, n], f_i(x) > f_i(y)$. A Pareto optimal solution is not dominated by any solution, and the set composed of all Pareto optimal solutions is referred to as the Pareto optimal solution set. One metric to evaluate the quality of a Pareto optimal solution set is hypervolume, which is illustrated in Figure 9. The hypervolume of a set is the volume of the space that is dominated by the solution in the set. When calculating the hypervolume of a set, we need to choose a reference point. When reference points are fixed, a Pareto solution set with a larger hypervolume is considered superior.

**Definition F.1** (Hypervolume metric). Let $P$ be a Pareto front approximation in an $n$-dimensional objective space and contain $N$ solutions. Let $r_0 \in R^m$ be the reference point. Then, the hypervolume metric is defined as:

$$\mathcal{H}(P, r_0) = \int_{R^n} \mathbb{1}_{H(P,r_0)(z)dz}$$

, where $H(P, r_0) = \{z \in Z | \exists 1 \leq i \leq |P| : r_0 \leq z \leq P(i)\}$. $P(i)$ is the i-th solution in $P$, $\leq$ is the relation operator of objective dominance, and $\mathbb{1}_{H(P,r_0)}$ is a Dirac delta function that equals 1 if $z \in H(P, r_0)$ and 0 otherwise.

### F.8 Contribution of Our Work to AI Community

**Advancing AI Chips** Our work directly contributes to the advancement of AI chips, such as NVIDIA's GPUs and Google's TPUs, by introducing an innovative optimization framework for the design of high-speed, area-efficient, and energy-efficient computing circuits. Note that NVIDIA's AI researchers have integrated AI-designed adders into their H100 chip [5], demonstrating the potential of our AI-based approach for advancing AI chips. The ability to optimize AI chips is crucial for addressing the ever-growing computational demands of modern AI systems, ensuring their scalability, efficiency, and sustainability.

**A Novel and Broadly Applicable Genetic Evolution Algorithm** Our work introduces a sequential mutation operator and a multi-granularity crossover operator that leverages a grid-based genetic solution representation to facilitate efficient and diverse exploration of large search spaces. This approach presents a broadly applicable framework suitable for addressing a wide range of search problems. Moreover, we propose a model-based cascade ranking approach, which efficiently and accurately selects high-performing solutions from a large pool of generated candidates. These contributions offer a versatile and robust methodology for tackling complex optimization problems.

**Identifying the Limitations of a Commonly-Used RL Formulation for Combinatorial Optimization** The existing RL formulation for computing circuit optimization adheres to a widely adopted paradigm in neural combinatorial optimization [6, 7, 8], commonly referred to as the "learn-to-improve" framework. In this paradigm, the state is defined as a candidate solution, the action represents a local modification to the solution, and the reward is based on the performance improvement achieved. This paper theoretically and empirically demonstrates that the RL-based formulation tends to converge to local optima, primarily due to deceptive reward signals and incrementally localized actions. These findings provide valuable insights for developing more robust and effective methods applicable to a broad class of neural combinatorial optimization problems.

## G Licenses

We credit the following open-source code and data used in this paper. We will also open-source our code once the paper is accepted.

**Environment**

1. OpenRoadFlowScripts BSD 3-Clause License
2. OpenRoad BSD 3-Clause License
3. Yosys ISC License
4. EasyMAC No License

**Algorithms**

1. GOMIL No License
2. MBPO MIT License
3. NovelD Creative Commons Public Licenses

# H   MORE RESULTS

## H.1   MORE RESULTS OF MAIN EVALUATION

The details about the hypervolume of MUTE and other baselines can be found at Table 6 and Table 7. Table 6 records the hypervolumes of the method on four multipliers with different bit-widths based on the And-Gate, showing that our method has the greatest improvement. Table 7 shows our improvements on the multipliers based on Booth-encode.

Table 6: We record the hypervolume of multipliers based on And-Gate.The results demonstrate that MUTE has the maximum hypervolume on each circuit design task.

| | 8-bit And | | 16-bit And | | 32-bit And | | 64-bit And | |
|---|---|---|---|---|---|---|---|---|
| Methods | HyperVolume | Improvement (%) | HyperVolume | Improvement (%) | HyperVolume | Improvement (%) | HyperVolume | Improvement (%) |
| Wallace | 149.94 | NA | 332.91 | NA | 1685.61 | NA | 13870.80 | NA |
| GOMIL | 153.02 | 2.05% | 394.25 | 18.43% | 3304.67 | 96.05% | 16628.51 | 19.88% |
| RL-MUL | 160.84 | 7.27% | 470.78 | 41.41% | 5329.71 | 216.19% | 25311.45 | 82.48% |
| AdaReset | 168.34 | 12.27% | 473.20 | 42.14% | 5768.29 | 242.21% | 32827.17 | 136.66% |
| HAVE | 179.49 | 19.71% | 504.94 | 51.67% | 5822.03 | 245.40% | 33030.52 | 138.13% |
| MUTE (Ours) | **189.68** | **26.50%** | **622.55** | **87.00%** | **6461.65** | **283.34%** | **36419.85** | **162.56%** |

Table 7: We record the hypervolume of multipliers based on Booth-encode. The results demonstrate that MUTE has the maximum hypervolume on each circuit design task.

| | 8-bit Booth | | 16-bit Booth | | 32-bit Booth | | 64-bit Booth | |
|---|---|---|---|---|---|---|---|---|
| Methods | HyperVolume | Improvement (%) | HyperVolume | Improvement (%) | HyperVolume | Improvement (%) | HyperVolume | Improvement (%) |
| Wallace | 304.86 | NA | 625.70 | NA | 4045.40 | NA | 13184.57 | NA |
| GOMIL | 314.22 | 3.07% | 773.43 | 23.61% | 3686.76 | -8.87% | 11456.09 | -13.11% |
| RL-MUL | 339.72 | 11.43% | 897.02 | 43.36% | 6090.67 | 50.56% | 19341.59 | 46.70% |
| AdaReset | 339.72 | 11.43% | 910.00 | 45.44% | 6970.98 | 72.32% | 23946.19 | 81.62% |
| HAVE | 339.72 | 11.43% | 975.94 | 55.98% | 7452.70 | 84.23% | 25910.38 | 96.52% |
| MUTE (Ours) | **366.63** | **20.26%** | **1060.86** | **69.55%** | **8057.07** | **99.17%** | **29441.15** | **123.30%** |

Table 8: The runtime of MUTE is comparable to or shorter than that of the recent state-of-the-art HAVE, while MUTE significantly improves hypervolume.

| | 16-bit And | | | 32-bit And | | |
|---|---|---|---|---|---|---|
| **RL/ERL Method** | RunTime (hours) | HV ↑ | Iterations | RunTime (hours) | HV ↑ | Iterations |
| MUTE (Ours) | 17.33 | 622.55 | 400 | 37.11 | 6461.65 | 400 |
| RL-MUL | 14.75 | 470.78 | 400 | 31.17 | 5329.71 | 400 |
| ParetoReset | 15.37 | 473.2 | 400 | 31.37 | 5606 | 400 |
| HAVE | 20.33 | 505.83 | 400 | 37.27 | 5822.03 | 400 |
| VEB-RL | 33.17 | 485 | 400 | 64.97 | 5402 | 400 |
| MBPO | 28.4 | 491.53 | 400 | 51.73 | 4978.11 | 400 |
| **EA Method** | | | | | | |
| MFEA | 10.7 | 473.39 | 400 | 19.7 | 5478.03 | 400 |
| MBBO | 12.7 | 473.36 | 400 | 35.1 | 5445.18 | 400 |

## H.2   RUNTIME COMPARISON OF MUTE WITH BASELINES

The results in Table 8 indicate that the runtime of our method is comparable to or shorter than that of the recent state-of-the-art HAVE (Wang et al., 2024g), while our method significantly improves the hypervolume of found Pareto points.

## H.3   MORE RESULTS OF COMPARISON WITH RL METHODS

Figure 10 illustrates the Pareto frontier of our MUTE and all RL-based methods. Moreover, we provide the results of hypervolume on 16-bit Booth and 32-bit Booth in Table 9. Through the table and figure, we can observe that MUTE outperforms other RL methods comprehensively, achieving the smallest area and delay.

Table 9: Results of comparison with RL methods on 16-bit Booth and 32-bit Booth.

| | 16-bit Booth | | 32-bit Booth | |
|---|---|---|---|---|
| Methods | HyperVolume | Improvement (%) | HyperVolume | Improvement (%) |
| Wallace | 625.70 | NA | 4045.40 | NA |
| RL-MUL | 897.02 | 43.36 | 6090.67 | 50.56 |
| NoveID | 873.31 | 39.57 | 6436.80 | 59.11 |
| PD-MORL | 871.02 | 39.21 | 6639.44 | 64.12 |
| DDQN | 894.74 | 43.00 | 6504.40 | 60.79 |
| MBPO | 942.20 | 50.58 | 6056.30 | 49.71 |
| RL-EA | 910.00 | 45.44 | 6970.98 | 72.32 |
| MBPO-EA | 932.86 | 49.09 | 6772.67 | 67.42 |
| MUTE (Ours) | **1060.86** | **69.55** | **8057.07** | **99.17** |

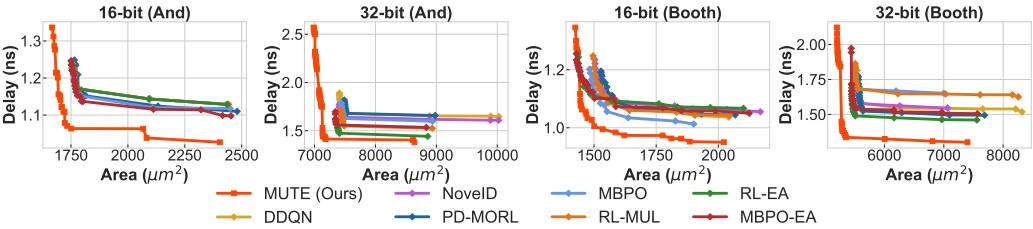

Figure 10: The results demonstrate that multipliers optimized by MUTE consistently and significantly outperform designs produced by all RL-based methods in terms of Pareto-dominance across four multiplier design problems.

## H.4 MORE RESULTS OF GENERALIZATION

Table 11 shows the hypervolume of PE arrays designed by MUTE and other baselines. MUTE achieves the highest hypervolume across all circuit designs.

## H.5 MORE ABLATION STUDY

**The Importance of Our Genetic Evolution Formulation** Although we have compared our MUTE with three specifically designed RL methods for computing circuits optimization, i.e., RL-MUL, AdaReset, and HAVE, the three methods are all based on the deep Q-network (DQN) algorithm (Mnih et al., 2015), which is a classical RL method. To further demonstrate the superiority of our formulation over the existing RL formulation, we further apply five advanced RL methods to multiplier design tasks. Specifically, we compare MUTE with four advanced RL methods, including NovelD (Zhang et al., 2021), PD-MORL (Basaklar et al., 2022), DDQN (Van Hasselt et al., 2016), and MBPO (Janner et al., 2019), and an evolutionary RL method, i.e., VEB-RL (Li et al., 2024b).

Table 10: MUTE significantly outperforms advanced (evolutionary) RL methods.

| | 16-bit And | | 32-bit And | |
|---|---|---|---|---|
| Methods | HyperVolume ↑ | Improvement(%) ↑ | HyperVolume ↑ | Improvement(%) ↑ |
| Wallace | 332.91 | NA | 1685.61 | NA |
| Specifically Designed RL Methods | | | | |
| RL-MUL | 470.78 | 41.41 | 5329.71 | 216.19 |
| AdaReset | 473.20 | 42.14 | 5768.29 | 242.21 |
| HAVE | 504.94 | 51.67 | 5822.03 | 245.40 |
| Advanced Standard RL Methods | | | | |
| NoveID | 473.20 | 42.14 | 4953.97 | 193.90 |
| PD-MORL | 485.03 | 45.69 | 4665.43 | 176.78 |
| DDQN | 473.20 | 42.14 | 4773.51 | 183.19 |
| MBPO | 491.53 | 47.65 | 4978.11 | 195.33 |
| SOTA Evolutionary RL Method | | | | |
| VEB-RL | 485.00 | 45.69 | 5402.00 | 220.48 |
| Our Genetic Evolution Formulation | | | | |
| MUTE (Ours) | **622.55** | **87.00** | **6461.65** | **283.34** |

The results in Table 10 suggest the following key conclusions. (1) MUTE significantly outperforms these advanced (evolutionary) RL methods, demonstrating the superiority of our proposed circuit genetic evolution formulation. (2) Advanced RL methods do not consistently and significantly outperform DQN-based circuit optimization methods, i.e., RL-MUL, AdaReset, and HAVE. This implies that the multiplier optimization task diverges significantly from standard RL benchmarks, such as Mujoco control (Todorov et al., 2012), due to its unique challenges.

Table 11: We record the hypervolume of PE arrays across four PE array design problems. The results demonstrate that MUTE has the maximum hypervolume on each circuit design task.

| | 16-bit And | | 32-bit And | | 16-bit Booth | | 32-bit Booth | |
|---|---|---|---|---|---|---|---|---|
| Methods | HyperVolume | Improvement (%) | HyperVolume | Improvement (%) | HyperVolume | Improvement(%) | HyperVolume | Improvement(%) |
| Wallace | 73073.23 | NA | 329263.00 | NA | 156874.00 | NA | 692705.30 | NA |
| GOMIL | 84820.07 | 16.08% | 627041.10 | 90.44% | 174168.70 | 11.02% | 628020.70 | -9.34% |
| RL-MUL | 103507.80 | 41.65% | 944998.90 | 187.00% | 219779.80 | 40.10% | 1049519.00 | 51.51% |
| AdaReset | 104073.20 | 42.42% | 1009747.00 | 206.67% | 230816.90 | 47.14% | 1237265.00 | 78.61% |
| HAVE | 108277.90 | 48.18% | 1019332.00 | 209.58% | 247851.40 | 57.99% | 1306568.00 | 88.62% |
| MUTE (Ours) | **143485.40** | **96.36%** | **1165307.00** | **253.91%** | **258221.60** | **64.60%** | **1388204.00** | **100.40%** |

