# OpenReview forum: "Computing Circuits Optimization via Model-Based Circuit Genetic Evolution"
_ICLR.cc/2025/Conference — ICLR 2025 Poster_

### Official Review · Reviewer_JnCS · 2024-11-03

**Soundness:** 3
**Presentation:** 4
**Contribution:** 3
**Rating:** 6
**Confidence:** 3

**Summary:**

This paper proposed a novel model-based circuit genetic evolution (MUTE) framework for optimizing computing circuits such as multipliers and adders by proposing a grid-based genetic representation of design solutions, which is true objective value rather than proxy rewards used in reinforcement learning (RL) frameworks. This work also proposes a multi-granularity genetic crossover operator to promote globally diverse exploration. Experiment results demonstrate MUTE’s superior ability to optimize the circuit to achieve excellent performance in terms of area and latency and is able to generalize to large-scale computation-intensive circuits.

**Strengths:**

• The paper is well-written and has both theoretical and experimental details to support its argument about RL's focus on local optimization.
• MUTE achieves relatively better local optimum compared to previous methods in empirical results.

**Weaknesses:**

• The paper argues that RL’s objective function is a proxy since it optimizes cumulative reward rather than the best reward achieved in the trajectory. This might be a significant problem in greedy RL algorithms such as DQN. There are other RL algorithms, such as PPO, that use entropy to promote exploration to escape local optimum. How does MUTE compare to this type of RL algorithm that promote exploration during optimization?
• The runtime for MUTE is longer than that of previous EA methods and some RL methods.

**Questions:**

Genetic Evolution Algorithms (GA) are established algorithms developed years ago. Is MUTE the first GA approach applied to optimize computing circuits? Are there any other global optimization algorithms, such as simulated annealing or other types of EA algorithms, used in this problem? The majority of the paper seems to narrow the related work to only EA and RL, which might not be the whole picture of this problem. I am curious about the advantages and novelty MUTE has over other traditional global optimization algorithms.

---

> ### Comment · Reviewer_JnCS · 2024-11-25
>
> I appreciate the author's response. Here are my comments on your assessment of RL's ability to find the global optimum. Reinforcement learning is a learning framework focusing on generalization (i.e., training on certain circuit design tasks and generalizing to other unseen design tasks within a few tries or one try). That's the reason RL focuses on maximizing accumulated reward instead of the best reward because RL does not want to overfit certain designs. Of course, it's highly likely gonna find a suboptimal solution compared to a dedicated optimizer that optimizes this design from scratch. Instead of focusing on 20%-50% hypervolume gain, the reason for using RL is to greatly save the design time during inference for unseen design which makes the RL scalable to different tasks since it does not need to optimize everything from scratch. Thus, my questions are: do the RL baselines in your paper all train from scratch for different tasks, or do they just perform inference, which costs them over 10 hours to complete? If it's the first way, I believe it's an incorrect way to use RL by using it as an optimizer, which greatly limits its potential as a learning framework. If it's the second way, I wonder why it takes over 10 hours for inference.

---

> > ### Author Response · Authors · 2024-11-25
> > **Further response to Reviewer JnCS**
> >
> > Dear Reviewer JnCS,
> >
> > Thank you very much for your thoughtful review and the time you have devoted to evaluating our rebuttal. We deeply appreciate your further insightful comments on the assessment of RL's ability. We respond to the comments in detail as follows and sincerely hope that our further response could properly address your concerns. If so, we would deeply appreciate it if you could raise your score. If not, please let us know your further concerns, and we will continue actively responding to your comments and improving our submission.
> >
> > > **1. Do the RL baselines in your paper all train from scratch for different tasks, or do they just perform inference?**
> >
> > In our paper, we follow previous RL-based computing circuits optimization work [1,2,3,4] to **train RL agents from scratch** for different computing circuits design tasks with different input bit widths.
> >
> > > **2. If it's the first way, I believe it's an incorrect way to use RL by using it as an optimizer, which greatly limits its potential as a learning framework.**
> >
> > We acknowledge that RL is a powerful learning framework with strong generalization capabilities for unseen states within a given environment. This is also a major reason why many prior works [1,2,3,4] have adopted RL to address computing circuit design tasks, demonstrating significant advancements over traditional design methods and integer programming-based optimization methods.
> >
> > However, we emphasize that **each computing circuit design task indeed corresponds to an environment with distinct state and action spaces in the previously proposed RL formulation**. These variations arise from differences in the input bit widths of the computing circuits like multipliers, leading to different state and action spaces. In this scenario, each circuit design task is analogous to a game in Atari. Consequently, training a single RL agent to handle various circuit design tasks mirrors the challenge of developing a unified RL agent capable of playing multiple Atari games, a task known to be inherently challenging.  Nevertheless, **developing a unified RL agent capable of addressing diverse computing circuit design tasks holds the potential to unlock the strong generalization ability of RL, which can be a compelling and critical avenue for future research**.
> >
> > [1] Rajarshi Roy, et al. PrefixRL: Optimization of Parallel Prefix Circuits using Deep Reinforcement Learning. DAC 2021.
> >
> > [2] Dongsheng Zuo, et al. RL-MUL: Multiplier Design Optimization with Deep Reinforcement Learning. DAC 2023.
> >
> > [3] Zhihai Wang, et al. A Hierarchical Adaptive Multi-Task Reinforcement Learning Framework for Multiplier Circuit Design. ICML 2024.
> >
> > [4] Yao Lai, et al.  Scalable and Effective Arithmetic Tree Generation for Adder and Multiplier Designs. NeurIPS 2024.

---

> > > ### Comment · Reviewer_JnCS · 2024-11-25
> > >
> > > Thanks for your response. Overall, this paper has an excellent presentation with a lot of details in theory and experiment. I have raised the presentation score to 4. After consideration, I decided to keep my overall score to 6 since I feel like the algorithm the author proposed is yet another optimizer without fundamental distinction from previous optimization work. I do recommend this paper to be accepted with all of its efforts on proof and experiments. Hope it gets accepted. Good Luck!

---

> > > > ### Author Response · Authors · 2024-11-28
> > > > **Further explanation on distinction from previous work (1/2)**
> > > >
> > > > # Further Response to Reviewer JnCS (1/2)
> > > > Dear Reviewer JnCS,
> > > >
> > > > Thank you sincerely for your prompt feedback and the invaluable time you have dedicated to reviewing our work! Your insights are crucial in helping us improve the quality of our submission! We are deeply grateful for your recognition of our efforts in the presentation, theoretical framework, and experiments, as well as your constructive suggestions for enhancing our work.
> > > >
> > > > To further improve our work, **we have explained in detail on the fundamental distinctions between our work and previous optimization work**. We sincerely hope that our further explanation could properly address your concerns. If not, we would be most grateful if you could share your additional concerns, and we are fully committed to continuing our efforts to refine and improve our submission based on your valuable feedback.
> > > >
> > > > > **1. I feel like the algorithm the author proposed is yet another optimizer without fundamental distinction from previous optimization work**
> > > >
> > > > **Introduction of the computing circuit optimization problem** For a computing circuit designed to perform a specific arithmetic function, such as an 8-bit multiplier, there are numerous structural designs that can meet the functional requirements. Each of these structural designs is referred to as a design solution. Notably, different circuit design structures can exhibit significant variations in performance metrics, such as area and delay.
> > > >
> > > > The computing circuit optimization problem **aims to seek the optimal design structure** that delivers the **best performance** for a **specific arithmetic function**, such as an 8-bit multiplier, by exploring a vast design solution space. Fundamentally, **this problem is a search task focused on seeking the optimal solution within a vast discrete solution space**. In this context, each solution represents a specific circuit design structure tailored to the given arithmetic function, with its performance serving as the optimization objective.
> > > >
> > > > **Previous computing circuit optimization work** The computing circuit optimization problem, which seeks the optimal solution within a large discrete solution space, is akin to other well-known search problems, such as neural architecture search and combinatorial optimization. RL has proven to be a powerful framework for discovering high-performing solutions in both neural architecture search [1, 2] and combinatorial optimization [3, 4]. Similarly, circuit design involves analogous search tasks, such as logic synthesis sequence optimization and macro placement. In these contexts, RL has been successfully employed to identify high-performance logic synthesis sequences [5, 6] and optimize macro placements [7, 8] for a specific circuit. Consequently, prior studies [9, 10, 11, 12] have proposed leveraging RL to search for high-performing design solutions for a given computing circuit, such as an 8-bit multiplier. Overall, **this body of work highlights that RL is a strong approach for addressing the computing circuit optimization problem**.

---

> > > > > ### Author Response · Authors · 2024-11-28
> > > > > **Further explanation on distinction from previous work (2/2)**
> > > > >
> > > > > # Further Response to Reviewer JnCS (2/2)
> > > > >
> > > > > **Fundamental distinction from previous optimization work**
> > > > >
> > > > > (1) **A novel genetic formulation compared to previous RL formulation** To the best of our knowledge, MUTE is the first to reformulate the computing circuit optimization problem as a genetic evolution problem by proposing a grid-based genetic representation for circuit designs. This grid-based genetic formulation eliminates the reliance on misleading rewards and enables globally diverse exploration.
> > > > >
> > > > > (2) **Novel genetic variation operators based on the grid-based genetic representation** MUTE designs a policy-guided **sequential mutation operator**, which **avoids myopic exploration** of the solution space by performing local modifications sequentially guided by a policy. MUTE designs a **multi-granularity crossover operator**, which effectively escapes local optima by recombining design substructures at varying column ranges between two grid-based genetic solutions.
> > > > >
> > > > > (3) **Model-based cascade ranking** MUTE proposes a model-based cascade ranking strategy that enables the **efficient** and **accurate** selection of high-performing solutions from a large pool of candidates. This approach **significantly enhances the effectiveness** of solution evaluation while **maintaining computational efficiency**.
> > > > >
> > > > > [1] Yesmina Jaafra, et al. Reinforcement Learning for Neural Architecture Search: A Review. Image and Vision Computing 2019.
> > > > >
> > > > > [2] Barret Zoph, et al. Neural Architecture Search with Reinforcement Learning. ICLR 2017.
> > > > >
> > > > > [3] Nina Mazyavkina, et al. Reinforcement Learning for Combinatorial Optimization: A Survey. Computers & Operations Research 2021.
> > > > >
> > > > > [4] Irwan Bello, et al. Neural Combinatorial Optimization with Reinforcement Learning. ICLR 2017.
> > > > >
> > > > > [5] Abdelrahman Hosny, et al. DRiLLS: Deep Reinforcement Learning for Logic Synthesis. ASPDAC 2020.
> > > > >
> > > > > [6] Jianyong Yuan, et al. EasySO: Exploration-enhanced Reinforcement Learning for Logic Synthesis Sequence Optimization and a Comprehensive RL Environment. ICCAD 2023.
> > > > >
> > > > > [7] Yao Lai, et al. MaskPlace: Fast Chip Placement via Reinforced Visual Representation Learning. NeurIPS 2022.
> > > > >
> > > > > [8] Zijie Geng, et al. Reinforcement Learning within Tree Search for Fast Macro Placement. ICML 2024.
> > > > >
> > > > > [9] Rajarshi Roy, et al. PrefixRL: Optimization of Parallel Prefix Circuits using Deep Reinforcement Learning. DAC 2021.
> > > > >
> > > > > [10] Dongsheng Zuo, et al. RL-MUL: Multiplier Design Optimization with Deep Reinforcement Learning. DAC 2023.
> > > > >
> > > > > [11] Zhihai Wang, et al. A Hierarchical Adaptive Multi-Task Reinforcement Learning Framework for Multiplier Circuit Design. ICML 2024.
> > > > >
> > > > > [12] Yao Lai, et al.  Scalable and Effective Arithmetic Tree Generation for Adder and Multiplier Designs. NeurIPS 2024.

---

> > > > > > ### Author Response · Authors · 2024-11-30
> > > > > > **Sincere Gratitude for Your Thoughtful Review and Feedback**
> > > > > >
> > > > > > Dear Reviewer JnCS,
> > > > > >
> > > > > > We would like to express our heartfelt gratitude for your kind support, prompt feedback, and encouraging words regarding our work! Your valuable insights have played a crucial role in helping us improve the quality of our submission, for which we are truly grateful!
> > > > > >
> > > > > > With sincere dedication, we have carefully addressed your remaining concerns on "**distinction from previous optimization work**", as detailed in our preceding responses—in response to your invaluable comments. Moreover, we are deeply grateful that both reviewers, cbXg and UJgx, have kindly recognized the novelty and contributions of this work. We sincerely hope that their thoughtful acknowledgment could assist in addressing the remaining concerns you might have. Specifically, Reviewer cbXg commented in strengths that "**The proposed framework is a fresh perspective on circuit design optimization**" and "**The paper makes a notable contribution to the field of circuit optimization**". Reviewer UJgx commented in strengths that "**novel application of genetic algorithms for HW optimization**".
> > > > > >
> > > > > > If, upon reflection, you find that our further response have properly addressed your remaining concerns, *we would be deeply grateful if you could consider revising your score to further support our paper*. Please be assured that we hold your expert judgment *in the highest regard*, and *we sincerely hope that this request does not cause any inconvenience or disturbance.*
> > > > > >
> > > > > > Regardless of your decision, we are sincerely grateful for the time and effort you have dedicated to reviewing our submission, and we are fully committed to continuing our efforts to improve our submission based on your valuable feedback.
> > > > > >
> > > > > > Best regards,
> > > > > >
> > > > > > Authors of #7706

---

### Official Review · Reviewer_cbXg · 2024-11-03

**Soundness:** 4
**Presentation:** 4
**Contribution:** 3
**Rating:** 8
**Confidence:** 4

**Summary:**

This paper tackles a fundamental challenge in circuit design by proposing an innovative approach called MUTE, a model-based circuit genetic evolution framework. The authors effectively identify limitations in current reinforcement learning (RL)-based methods, such as deceptive reward signals and limited action scope, which hinder achieving optimal circuit designs. MUTE addresses these gaps by introducing a grid-based genetic representation, allowing for a global search of design solutions that enhance performance across various circuits, including multipliers, adders, and multiply-accumulate circuits. Experimental results are impressive, with MUTE consistently outperforming state-of-the-art methods in area and delay optimization, achieving up to a 38% improvement in hypervolume metrics. Additionally, MUTE shows strong scalability to larger, more complex circuits, making it a highly promising direction for future research in circuit optimization.

**Strengths:**

- The authors effectively identify and address significant shortcomings in existing RL-based methods, specifically the issues of deceptive reward signals and localized actions, which often lead to suboptimal solutions in circuit design.
- The proposed framework, MUTE, is a fresh perspective on circuit design optimization. Specifically, MUTE's grid-based genetic representation allows for a more comprehensive and global search of optimal design solutions, providing a significant advantage over RL methods that may focus too narrowly on localized actions.
- The paper backs its claims with extensive empirical evidence, showing that MUTE outperforms state-of-the-art methods on fundamental computing circuits like multipliers, adders, and multiply-accumulate circuits.
- The evaluation section is comprehensive and detailed; the results are also promising. They demonstrate the effectiveness of the MUTE framework, with improvements in hypervolume metrics by up to 38% and clear and measurable advancements in area and delay optimization.
- Overall, the paper is very well written and flows nicely. Also, by introducing a framework that can Pareto-dominate existing and recent methods in terms of area efficiency and speed, the paper makes a notable contribution to the field of circuit optimization.

**Weaknesses:**

- The authors posit an assumption on deceptive reward signals—through the inconsistency between the RL and original optimization objectives. While a detailed theoretical proof is provided in the appendix, the oscillations shown in Figure 2 and Figure 6 are highly related to the search space and the scope of the underlying optimization problem. The authors are recommended to study the optimization objective landscape and the convergence into local optima in previous RL-based works, notably RL-MUL, AdaReset, and HAVE.
- The idea behind RL-guided mutation to apply strategic modifications to the circuit design is compelling. However, details about the learned Q-network and how it guides the mutation are missing in the paper, particularly in section 4.3. The authors should add more details about the Q-network model, learning, action space (i.e., what mutations are to be applied), and more importantly, how the evolutionary framework handles invalid circuit designs after mutations.
- The rationale behind leveraging two cascade models to rank the explored circuit designs needs to be further justified–specifically, why use two models instead of one ranking model that can be updated with real-world evaluation during the evolution process? Furthermore, details about the learned model to estimate the fitness and performance of the circuit design are missing. The authors are recommended to emphasize the fitness evaluation stage and running methods proposed in this paper by conducting an ablation study when using a single ranking model, a single adaptive ranking model (whose learning parameters can be updated by some real-world evaluations during the optimization process), the proposed two-stage ranking model, and ranking based on real-world evaluations only. The four strategies should be compared for performance estimation accuracy, Pareto ranking preserving, and evaluation time.

**Questions:**

- The authors assume that deceptive reward signals arise from inconsistencies between the RL and original optimization objectives, which may contribute to the oscillations observed in Figures 2 and 6. Could you elaborate on how this assumption impacts the results and whether analyzing the search space and optimization landscape in related works like RL-MUL, AdaReset, and HAVE might clarify these oscillations?
- The paper mentions an RL-guided mutation strategy for circuit design, yet details about the Q-network, including its structure, learning process, and action space, are limited. Could the authors provide more specifics on how the Q-network decides on mutations and manages invalid designs resulting from these mutations? Further explanation could clarify how the framework ensures robustness in the mutation process, especially in Section 4.3, where details on this aspect are limited.
- Why do the authors choose a two-stage cascade ranking model instead of a single, adaptive ranking model that could be continuously updated with real-world evaluations? Could you justify this choice by explaining the specific benefits of two models over one in the context of fitness evaluation and performance estimation? An ablation study is recommended to compare the proposed two-stage ranking model with a single ranking model, an adaptive ranking model, and real-world evaluation-based ranking. How would each strategy affect performance estimation accuracy, Pareto ranking preservation, and evaluation time?

---

> ### Comment · Reviewer_cbXg · 2024-11-24
>
> I greatly appreciate the authors' comprehensive and well-articulated responses, which have successfully addressed my concerns. I strongly encourage the authors to integrate these clarifications into the revised manuscript to further strengthen its presentation. Sharing the codebase would also be a valuable addition, promoting transparency and enabling reproducibility (although I've checked the codebase link shared in the responses to Reviewer UJgx, I strongly recommend the authors to add the link to their revised paper as well). Considering the significance of the paper's contributions and findings, I am happy to revise my score and recommend the paper for acceptance.

---

### Official Review · Reviewer_UJgx · 2024-11-03

**Soundness:** 3
**Presentation:** 4
**Contribution:** 2
**Rating:** 6
**Confidence:** 3

**Summary:**

Paper presents a genetic algorithm based optimization framework (MUTE) to finding more hardware efficient multipliers and adders with respect to area and latency. The main degrees of freedom are the placement of the compressor blocks in the compressor trees. The results clearly show that their framework finds more optimal pareto fronts compared to existing RL based optimization approaches and standard synthesis techniques.

**Strengths:**

1)  clear problem statement and objectives, with a lot of discourse on RL related work
2)  good discussion of the theory regarding multi-variable optimization and pareto fronts /hyper volume metrics
3)  nice coverage of the limitations of pure RL approaches
4)  sound results including lots of experiments and ablation studies
5)  novel application of genetic algorithms for HW optimization

**Weaknesses:**

1) limited discussion/exploration for hybrid RL/discrete optimization algorithms
2) lack of discussion about the larger contributions to AI/ML community (how this helps)
3) section 4.4 on the cascade ranking and sampling are important to the paper, but many details are deferred to the appendix.
4) Table 3 with the contribution for each component is a bit dubious - it suggests to me that the components may not be as important as stated in the paper since they all come close to the all inclusive framework  results
5) not sharing framework / code during evaluation phase (only conditionally based acceptance)

**Questions:**

1)  Have you considered hybrid RL / genetic optimization approaches?  What about other optimization algorithms (simulated annealing w/ predictive heuristics)?  Can you compare MUTE's performance with a simple simulated annealing + heuristics?
2)  Can you consider adding some context and tie-in to the larger benefits of this work to the AI/ML community? Either outside of circuit design, or even w/in circuit design, how it helps to advance the AI/ML field?
3)  From Table 3, there is a relatively small difference in hypervolume when removing individual components in the ablation study.  Can you provide a more detailed analysis of the trade-offs between performance gains and computational costs for each component? Can you also do the same for a simplified version of MUTE since it seems like this could be an interesting approach?
4)  Can you consider adding a discussion of the complexity and runtime tradeoffs in a new sub-section in the results section rather than the Appendix?

---

### Meta-Review · Area_Chair_ok4k · 2024-12-18

**Metareview:**

Optimizing computing circuits like multipliers and adders is challenging, with reinforcement learning approaches often converging to suboptimal designs due to deceptive reward signals and localized actions. To overcome this, the authors propose MUTE, a model-based circuit genetic evolution framework that uses a grid-based genetic representation and evaluates solutions with true objective values, avoiding proxy reward pitfalls. By incorporating a multi-granularity genetic crossover operator for diverse exploration, MUTE achieves Pareto-optimal circuit designs with superior area and delay efficiency, and its solutions generalize well to large-scale computation-intensive circuits.

Most reviewers acknowledge the novel angle, novel application, and throughout experiments. Concerns mostly lie in insufficient details and analysis. During the rebuttal period, the authors did a great job and addressed most concerns. We recommend an acceptance.

**Additional Comments On Reviewer Discussion:**

During the rebuttal period, the authors did a great job and addressed most concerns. We highly encourage the authors to incorporate these discussions accordingly.

---

### Decision · Program_Chairs · 2025-01-22

Accept (Poster)